# A multi-lock inhibitory mechanism for fine-tuning enzyme activities of the HECT family E3 ligases

Zhen Wang[1,3], Ziheng Liu[1,3], Xing Chen[1], Jingyu Li[1], Weiyi Yao[1], Shijing Huang[1], Aihong Gu[1], Qun-Ying Lei [1,2], Ying Mao[1] & Wenyu Wen [1]

HECT E3 ligases control the degradation and functioning of numerous oncogenic/tumor-suppressive factors and signaling proteins, and their activities must be tightly regulated to prevent cancers and other diseases. Here we show that the Nedd4 family HECT E3 WWP1 adopts an autoinhibited state, in which its multiple WW domains sequester HECT using a multi-lock mechanism. Removing WW2 or WW34 led to a partial activation of WWP1. The structure of fully inhibited WWP1 reveals that many WWP1 mutations identified in cancer patients result in a partially active state with increased E3 ligase activity, and the WWP1 mutants likely promote cell migration by enhancement of ΔNp63α degradation. We further demonstrate that WWP2 and Itch utilize a highly similar multi-lock autoinhibition mechanism as that utilized by WWP1, whereas Nedd4/4 L and Smurf2 utilize a slightly variant version. Overall, these results reveal versatile autoinhibitory mechanisms that fine-tune the ligase activities of the HECT family enzymes.

---

[1] Department of Neurosurgery, Huashan Hospital, State Key Laboratory of Medical Neurobiology and MOE Frontiers Center for Brain Science, Institutes of Biomedical Sciences, School of Basic Medical Sciences, Shanghai Medical College, Fudan University, Shanghai 200032, China. [2] Fudan University Shanghai Cancer Center and Cancer Metabolism Laboratory, Fudan University, Shanghai 200032, China. [3] These authors contributed equally: Zhen Wang, Ziheng Liu. Correspondence and requests for materials should be addressed to W.W. (email: wywen@fudan.edu.cn)

Ubiquitin ligases (E3s) determine the selectivity and the modification sites of target proteins[1–4] and are therefore key specificity factors in ubiquitin signaling and have attracted extensive attention as targets for therapeutic applications[5–8]. HECT (homologous to E6-AP carboxyl terminus)-type E3s play vital roles in diverse physiological processes, including cell cycle progression, cell proliferation, autophagy, and inflammation, and their dysregulation is closely correlated with human diseases such as cancers, immune disorders, and neurological diseases[9–14]. Each HECT-type E3 contains a characteristic HECT domain at its C terminus, which catalyzes the transfer of ubiquitin from E2 to itself and then to the specific substrate.

Based on their distinct N-terminal domains, HECT-type E3s can be further grouped into several subfamilies, and the Nedd4 family is the largest and best characterized of these subfamilies[7,9]. The Nedd4 family E3s have nine members in humans (WWP1/2, Nedd4/4 L, Smurf1/2, NEDL1/2, and Itch), which share a common N-terminal domain architecture comprised of a C2 domain and 2–4 WW domains that are responsible for subcellular localization and substrate recognition, respectively. Functionally, WWP1 has been postulated to function as an oncogenic factor by regulating the stability of several cancer-related proteins, such as p53, p63, KLF2, JunB, HER4, and KLF5, and its dysregulation has been implicated in cancers, infectious diseases, neurological diseases, and ageing[9,10,15]. WWP1/2 have recently been found to be essential for the acquisition of neuronal polarity[16]. *WWP1* mRNA and protein levels are upregulated in a substantial number of breast and prostate cancers[17,18], as well as in acute myeloid leukemia[19]. In addition, the *WWP1* gene has been found to be frequently mutated in human cancers[17,20], although the consequences of these mutations have not yet been elucidated.

The HECT domain adopts a bilobal structure, in which the E2-binding N-lobe is connected by a flexible hinge loop to the catalytic C-lobe[3]. Mutations in the hinge region that restrict subdomain rotation inhibit the ubiquitination of WWP1, suggesting that structural plasticity between the HECT N-lobe and the C-lobe is required for catalytic activity[21,22]. To prevent excessive ubiquitination of targets and self-destruction by autoubiquitination, HECT-type E3s normally adopt an inactive state characterized by intra- or inter-molecular interaction[7,23]. Several studies have shown that ubiquitination or N-terminal extension-dependent oligomerization of HECT regulate the activity of Rsp5 and HUWE1, respectively[24,25]. Increasing evidence has expanded our knowledge about the regulatory mechanisms utilized by the Nedd4 family E3s. The C2 domains in Smurf1/2 and Nedd4/4 L interact with the HECT domains to inhibit noncovalent ubiquitin binding and the transfer of ubiquitin to the E3 active site[26–29]. The autoinhibitory interaction can be released by the binding of adapter proteins or $Ca^{2+}$/phospholipids to C2 domains or the post-translational modification of the C2 and HECT domains[27–30]. Several recent studies on Itch, WWP2, and their *Drosophila* orthologue Su(dx) have shown that the WW2 domain and the linker region (referred to as L hereafter) between WW2 and WW3 synergistically interact with HECT to inhibit ligase activity by occupying the noncovalent ubiquitin binding site and restricting the flexibility of the two lobes[31–33]. Such WW2L-mediated autoinhibition can be released by binding to three PY-bearing adaptor Ndfip1 or JNK1-mediated phosphorylation sites through the WW domains or by the phosphorylation of L. However, many intriguing questions remain. Does a common regulatory mechanism exist for all HECT E3s or for a subgroup of HECT E3s? Are the characteristic WW domains in combination with the L region(s) involved in ligase regulation in other Nedd4 family members? Is there another regulation site(s) on the HECT domain that may provide new hotspots for therapeutic interventions?

In this study, we solved the crystal structures of WWP1 in its fully inactive (2L34HECT) and partially active (L34HECT) states (Fig. 1a). Detailed structural analysis shows that WW2, L, and WW4 are organized into a headset architecture, in which the WW2 and WW4 domains are bound to bilateral sites within the N-lobe, and L forms a kinked α-helix that is tucked into the cleft between the N- and C-lobes of HECT. WW2 and L of WWP1 interact with HECT in the same mode as does WWP2 and Itch[31,32]. Interestingly, the N-terminal extension of HECT occupies the canonical PY motif binding site in the WW4 domain. Further biochemical experiments indicated that WW2 and WW4 stabilize both termini of L. This extensive bilateral interaction kept WWP1 in a fully inactive state by preventing ubiquitin transfer from E2. Cancer-related mutations in the WW2L34/HECT interface enhanced WWP1 ligase activity, which likely promoted cell migration through increased ΔNp63α ubiquitination and degradation. Finally, we show that this multi-lock regulation mechanism is conserved in WWP2 and Itch, whereas in Nedd4/4 L and Smurf2, a variant version of the multi-lock autoinhibition mode is utilized.

## Results

**WW234 and L lock WWP1 in an inactive state**. The full-length (FL) WWP1 protein adopts the canonical domain organization typical of the Nedd4 family, which contains four WW domains (Fig. 1a) and a linker region (L) that connects WW2 and WW3, which has been suggested to be responsible for WWP1 autoinhibition[31,32]. Consistent with previous findings, FL WWP1 was kept in a closed conformation that exhibited low ligase activity. Deletion of the C2 domain (12L34HECT) or C2 together with WW1 (2L34HECT) had little impact on the ligase activity (Fig. 1a, b and Supplementary Table 1). To determine which domain(s) is responsible for the autoinhibition, we generated a series of additional truncation or deletion mutants, and tested their ligase activities by using an autoubiquitination assay (Fig. 1a, b). Deletion of L (234HECT) caused a striking increase in WWP1 autoubiquitination, indicating that L indeed has a dominant role in ligase regulation. However, although it is essential, L alone is not sufficient for a complete HECT inhibition, as L-HECT exhibited a comparable ligase activity to that of the isolated HECT domain. In contrast, the presence of the adjacent WW domain(s) together with L (e.g., 2LHECT or L34HECT) led to dramatic decreases in WWP1 autoubiquitination, suggesting that both WW2 and WW34 are responsible for ligase regulation. Both 2LHECT and L34HECT mutants possess slightly elevated ligase activity compared to 2L34HECT, indicating that they adopt a partially active state.

In line with the above data, the GST pull-down assay showed that the WW domain region (WW12L34, hereafter referred to as WW) but not the C2 domain could form a stable complex with HECT (Supplementary Fig. 1a). The finding that only GST-tagged WW2L or LWW34 could robustly pull-down thioredoxin (Trx)-tagged HECT, but not the GST-tagged isolated L, WW1, WW2, or WW34 (Fig. 1c and Supplementary Fig. 1b), indicated that WW domains function together with L to bind to HECT with high affinity. Moreover, the ligase activity of WWP1 HECT could be robustly inhibited by WW2L or LWW34, but not by the isolated WW2, WW34 and L, in a dose-dependent manner (Fig. 1d). Overall, the ubiquitination and pull-down data suggested that L is essential but not sufficient for WWP1 autoinhibition; WW2L or LWW34 alone can partially inhibit the ligase activity of WWP1, whereas WW2L34 together can keep WWP1 in the fully inactive state.

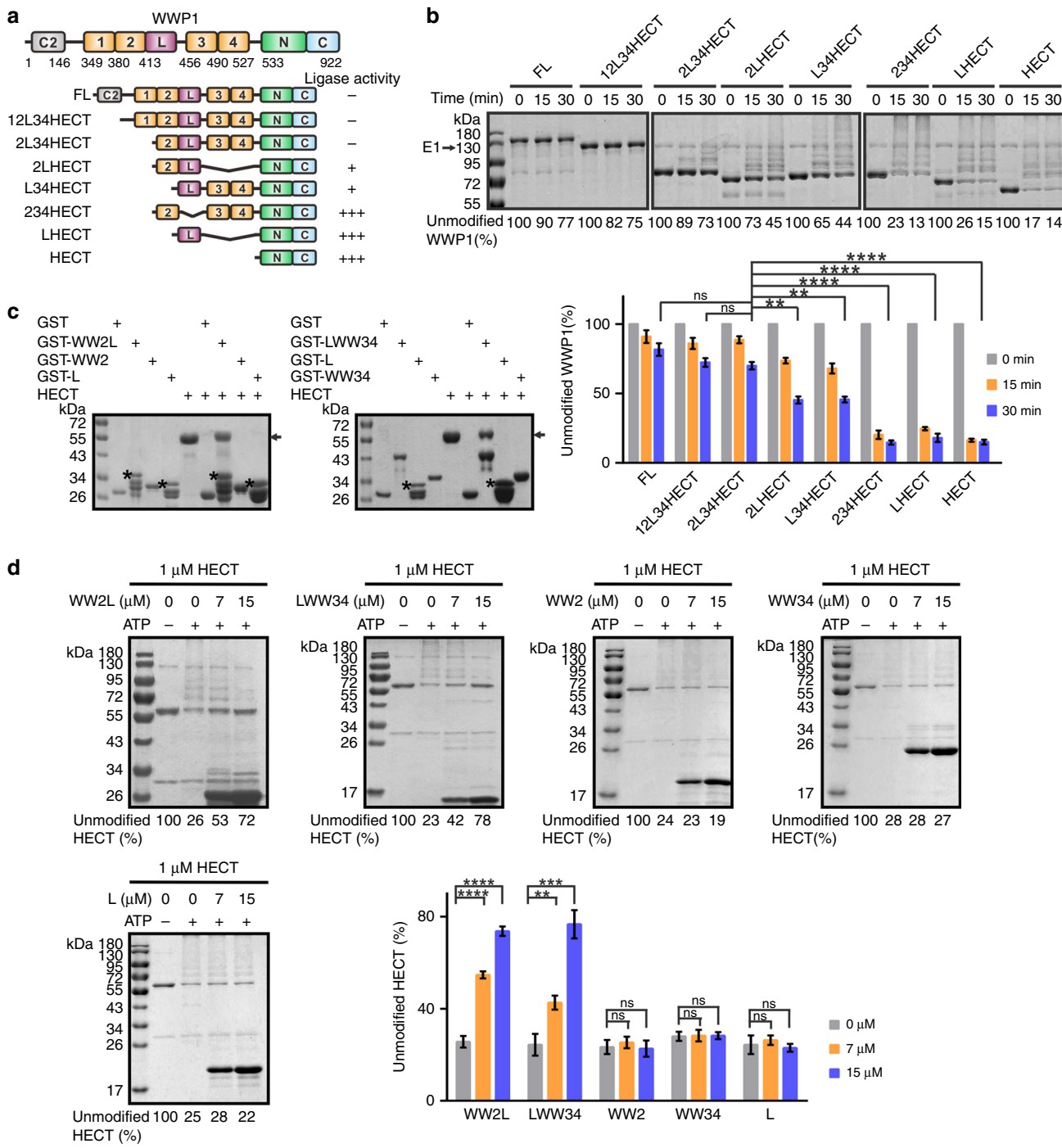

**Fig. 1** WW234 and L lock WWP1 in an inactive state. **a** Schematic of WWP1 domains showing an enzymatic activity summary derived from the autoubiquitination assay in **b**. **b** Autoubiquitination assay of Trx-tagged full-length (FL) WWP1 and various fragments. The weight markers belong to all separate gels. **c** GST pull-down assay of GST-tagged WW2L, WW2, L, LWW34, and WW34 with Trx-HECT. **d** WWP1 HECT autoubiquitination assay with different concentrations of Trx-WW2L, LWW34, Trx-WW2, Trx-WW34, and Trx-L. For the ubiquitination assay, the reactions were quenched after 15 min. Statistics for the enzymatic activities are shown below. Data are presented as the mean ± SD of triplicate experiments; ns, not significant, *$p < 0.05$, **$p < 0.01$, ***$p < 0.001$, and ****$p < 0.0001$ based on one-way analysis of variance (ANOVA) with Tukey's multiple comparison test. Source data are provided as a Source Data file

**Overall structures of the inactive and partially active WWP1.** To elucidate the molecular mechanism governing WWP1 auto-inhibition, we determined the crystal structures of WWP1 in its fully inactive (2L34HECT) and partially active (L34HECT) states at resolutions of 2.3 Å and 2.5 Å, respectively (Table 1). In the 2L34HECT crystal structure, WW2, WW4, HECT, and half of L (aa 413–427 and 432–446) are well-resolved, whereas the WW3

domain could not be modeled due to its poor electron density (Fig. 2a). The HECT domain in the closed form of WWP1 adopts a T-shape conformation, which is highly similar to that of the isolated WWP1 HECT domain structure[21]. WW2, L, and WW4 are organized into a headset architecture, with the WW2 and WW4 domains acting as the "right ear" (denoted as "Re") and the "left ear" (denoted as "Le"), respectively, when binding to bilateral

**Table 1 Data collection and refinement statistics**

| | WWP1 2L34HECT | WWP1 L34HECT |
|---|---|---|
| Data collection | | |
| Space group | P2₁ | P1 |
| Cell dimensions a, b, | 56.309, 45.444, | 60.019, |
| c (Å) | 116.545 | 59.370, 85.075 |
| α, β, γ (°) | 90.000, | 99.392, 92.292, |
| | 93.794, 90.000 | 108.751 |
| Wavelength (Å) | 0.9793 | 0.9793 |
| Resolution (Å) | 50.00–2.25 | 50.00–2.50 |
| | (2.29–2.25)* | (2.54–2.50)* |
| $R_{merge}$ (%) | 6.9 (35.7) | 10.8 (83.4) |
| Mean $I/\sigma$ | 16.3 (3.0) | 32.6 (2.3) |
| Completeness (%) | 97.7 (97.7) | 95.2 (92.3) |
| Redundancy | 3.5 (3.6) | 4.0 (3.4) |
| CC1/2 | (0.922) | (0.515) |
| Refinement | | |
| Resolution (Å) | 35.81–2.30 | 42.57–2.55 |
| No. reflections | 26023 | 33955 |
| $R_{work}$ /$R_{free}$ (%) | 21.95/27.05 | 20.81/25.43 |
| No. atoms | | |
| Protein | 3644 | 6327 |
| Water | 54 | 10 |
| B factors | | |
| Protein | 38.39 | 55.18 |
| Water | 37.41 | 48.26 |
| R.m.s deviations | | |
| Bond lengths (Å) | 0.007 | 0.008 |
| Bond angles (°) | 0.830 | 0.935 |

*Values in parentheses indicate the highest-resolution shell

sites of the N-lobe, whereas L functions as the "headband" of the headset (denoted as "H") by forming a kinked α-helix that is tucked into the cleft between the N- and C-lobes of HECT (Fig. 2a). Note that the binding modes of WW2 and L with HECT (at "Re" and "H" sites, respectively) in WWP1 2L34HECT are highly similar to those found in Itch and WWP2 (Supplementary Fig. 1c)[31,32]; however, the WW4-binding "Le" site in WWP1 HECT has not been reported.

In the L34HECT crystal structure, WW3, WW4, and the entire L region were well-resolved (Fig. 2b). It seems that the WW3 and WW4 domains are packed closely together, WW4 binds to the same "Le" site on HECT, and L forms a kinked α-helix that resides at the same "H" site that spans from the "Re" site to the "Le" site. Compared with 2L34HECT, WW4 and the N-terminal part of L are rotated ~35 and 25 degrees, respectively in L34HECT, and a few N-terminal residues (aa 413–416) in L were not observed (Fig. 2c and Supplementary Fig. 1d). Detailed structural analysis revealed that more crystal contacts exist in 2L34HECT (Supplementary Fig. 1e). E509 from WW4 of 2L34HECT forms a salt bridge with K531 from a symmetric molecule. The main chain of M414 from the N-terminal part of L of 2L34HECT forms a hydrogen bond with N745 of a symmetric molecule. All these crystal contacts do not exist in the L34HECT structure, implying that the rotation of WW4 and the N-terminal part of L in the two structures may arise from crystal contacts. Together with the GST pull-down results (Fig. 1c), the crystal structure results seem to suggest that the binding of the adjacent WW2 to the "Re" site may stabilize the N terminus of L. Presumably, the double lock (comprised of WW2 and WW4) better stabilizes the L/HECT interaction than the isolated ones (WW2 or WW4). In line with this analysis, WW2L34 pulled down more HECT than WW2L or LWW34 (Supplementary Fig. 1b). The E2–E3 transthiolation assay further suggests that, although L is essential for keeping WWP1 in its fully inhibited

state, the isolated L is not sufficient for locking HECT in the inactive state (Fig. 2d). Overall, the above biochemical and structural analyses indicate that the multi-lock autoinhibition mode that results from the binding of the WW2, L, and WW34 domains to the catalytic HECT domain maintains WWP1 in the fully inactive state.

**The WW2L34-HECT interface.** Detailed analysis of 2L34HECT suggested that the intramolecular packing is mainly driven by extensive hydrophobic and hydrogen bonding interactions (Fig. 3a). The hydrophobic Met627$^{HECT}$ interacts with a hydrophobic cluster formed by Trp387$^{WW2}$ and Phe420$^L$. Pro651$^{HECT}$ inserts into a small pocket formed by Trp409$^{WW2}$ and Arg396$^{WW2}$. These two core hydrophobic interactions physically anchor WW2 to HECT at the "Re" site. Hydrophobic Met434$^L$, Phe437$^L$, Tyr441$^L$, and Tyr443$^L$ interact with a hydrophobic cleft formed by Phe617$^{HECT}$, Phe673$^{HECT}$, and Met804$^{HECT}$ at the "H" site (Fig. 3a), whereas, at the "Le" site, the main force driving the WW4-HECT interaction is hydrogen bonding between His517$^{WW4}$ and Tyr543 in the N-terminal extension of HECT. Unexpectedly, the "IxY$^{543}$"-motif in the N-terminal extension of HECT occupied the canonical "NPxY"-motif binding site in WW4 (Supplementary Fig. 1f). Importantly, although the 35 degree rotation of WW4 and 25 degree rotation of L appear to be significant, the crucial WW4-HECT and L-HECT contacts in the 2L34HECT structure are preserved in L34HECT (Supplementary Fig. 1g). Our assumption is that the WW4-binding N-terminal extension of HECT is a flexible loop, which allows HECT-bound WW4 to rotate a certain extent upon crystal packing with a symmetric molecule in 2L34HECT; whereas the 25 degree rotation of the N-terminal part of L is the result of the missing WW2 domain and crystal contacts (Supplementary Fig. 1e). Thus, we assume that these crystal contact-induced conformational changes have no functional significance.

In agreement with the above structural analysis, a point mutation at the WW2-binding "Re" site (M627E$^{HECT}$) or the WW4-binding "Le" site (Y543A$^{HECT}$) on the HECT domain significantly weakened the interaction between WW and HECT, and double mutation of both sites (Y543A,M627E$^{HECT}$) eliminated binding (Fig. 3b). Similarly, a mutation at the "Re" site-binding surface on WW2 (W409A$^{WW2}$) or the "Le" site-binding pocket on WW4 (H517A$^{WW4}$) severely impaired the WW–HECT interaction, and the W409$^{WW2}$, H517A$^{WW4}$ double mutation disrupted the interaction (Fig. 3c). In addition, the F617E$^{HECT}$ mutation at the L-binding "H" site on HECT also eliminated WW-binding (Fig. 3d). Consistent with the GST pull-down results, the above mutations as well as others at the WW2L34-HECT packing interface in WW2 (W409A), L (F437A, Y441A, Y443A, and M447A), WW4 (E503A and H517A), or HECT (Y543A, Y543E, W549A, F617E, M627E, and M804, Q805A) all led to significantly elevated autoubiquitination of WWP1 (Fig. 3e and Supplementary Fig. 2).

We further validated the functional relevance of the WW4-HECT interaction in L34HECT. The point mutation at the WW4-HECT-binding surface (H517A$^{WW4}$ or Y543A$^{HECT}$) severely impaired the WW4-HECT interaction (Supplementary Fig. 2b). Meanwhile, mutations at the WW4-HECT packing interface in WW4 (E503A, H517A, and H517Y) or HECT (Y543A, Y543E, W549A) led to significantly elevated autoubiquitination of WWP1 L34HECT (Supplementary Fig. 2c), further demonstrating that the crystal packing-induced rotation of WW4 does not affect its inhibition in HECT activity.

Overall, the above structural and biochemical analyses highlight the critical role of WW2, L, and WW4 in maintaining a

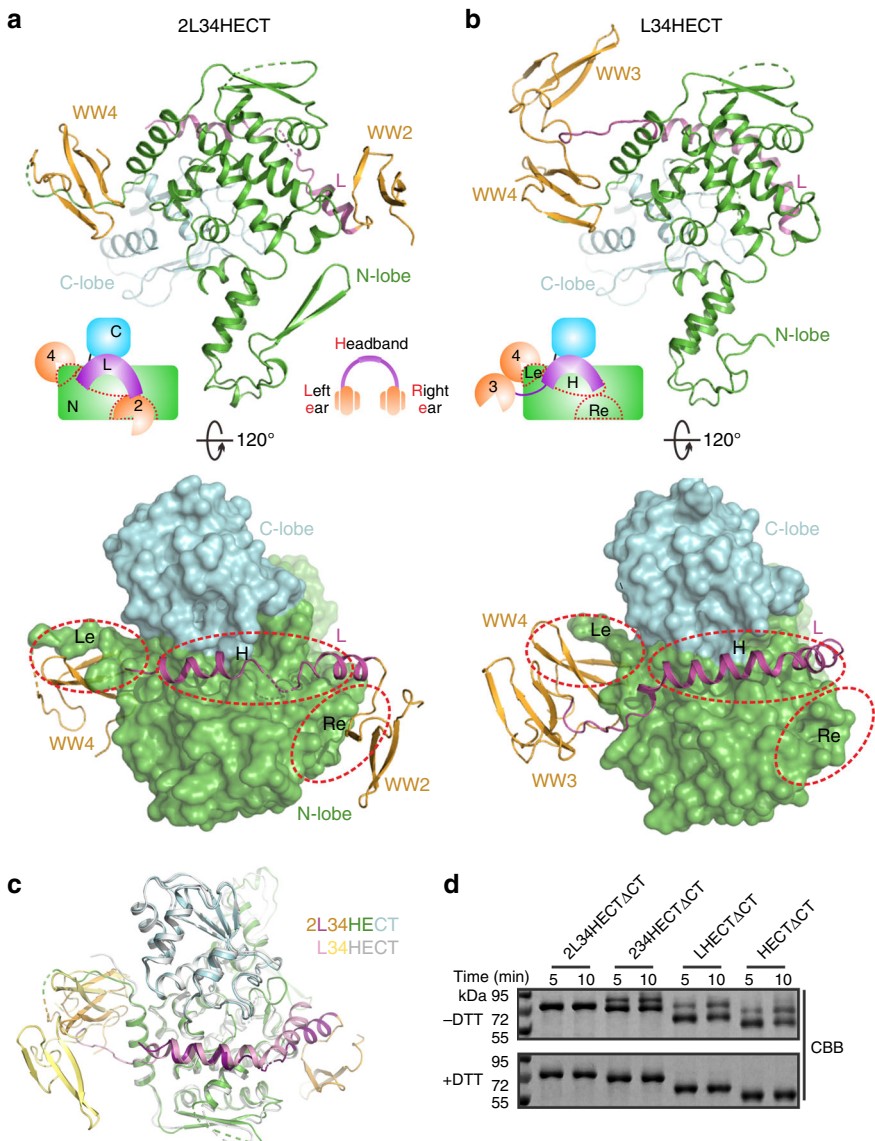

**Fig. 2** The overall structure of WWP1. **a–b** Ribbon and surface representation of WWP1 2L34HECT and L34HECT. **c** Structural comparison of WWP1 2L34HECT and L34HECT. **d** E2–E3 transthiolation assay of WWP1 C terminal five residue-deletion constructs (2L34HECTΔCT, 234HECTΔCT, L-HECTΔCT, and HECTΔCT). Source data are provided as a Source Data file

robust intramolecular WW–HECT interaction and thus keeping WWP1 in the completely inactive state.

**Cancer-related mutations of WWP1.** Accumulating evidence has demonstrated the vital role of WWP1 as an oncogenic factor that has been found to be frequently misregulated or mutated[9,10,15,20] in different human cancers, although the underlying mechanism is largely unclear. We surveyed the COSMIC cancer somatic mutation database[34] (http://cancer.sanger.ac.uk/cosmic) and found numerous mutations that occur in the *WWP1* gene. A significant number of these mutations are located in the WW2L34 and HECT domains (Supplementary Fig. 3a). The biochemical and structural information presented in this study allowed us to test the impact of some mutations that have been found in cancer patients in terms of their effect on the ligase activity of WWP1. For practical reasons, we chose to investigate the mutation sites that are located within the WW2L34-HECT packing interface (Fig. 4a). As expected, single missense mutations in the "Re" (P651A$^{HECT}$), "Le" (H517Y$^{WW4}$), or "H" (R427W$^{L}$ and S444L$^{L}$) sites all significantly increased

WWP1's ligase activity (Fig. 4b and Supplementary Fig. 2), further indicating that the activity of WWP1 needs to be tightly controlled in vivo.

**WWP1 promotes cell migration by regulating ΔNp63α turnover.** Taking advantage of the fact that the constitutively active WWP1 mutants described above are clearly understood from a mechanistic perspective (Figs. 3e and 4b), we asked whether and how the enzymatic activity of WWP1 might be involved in cancers by determining its involvement in cell proliferation and migration. To address this question, we examined the effect of several mutations (e.g., W549A$^{HECT}$ at the "Le" site and the cancer-related H517Y$^{WW4}$ and P651A$^{WW2}$ at the "Le" site and "Re" site, respectively) on cell migration using the human breast epithelial cell line MCF-10A as a model. In line with the in vitro ubiquitination data, all of these mutants dramatically promoted cell migration, as shown by wound-healing (Fig. 4c and Supplementary Fig. 3b) and cell proliferation (Fig. 4d) assays. Note that overexpression of the wild-type (WT) WWP1 only showed negligible effect on cell migration and proliferation

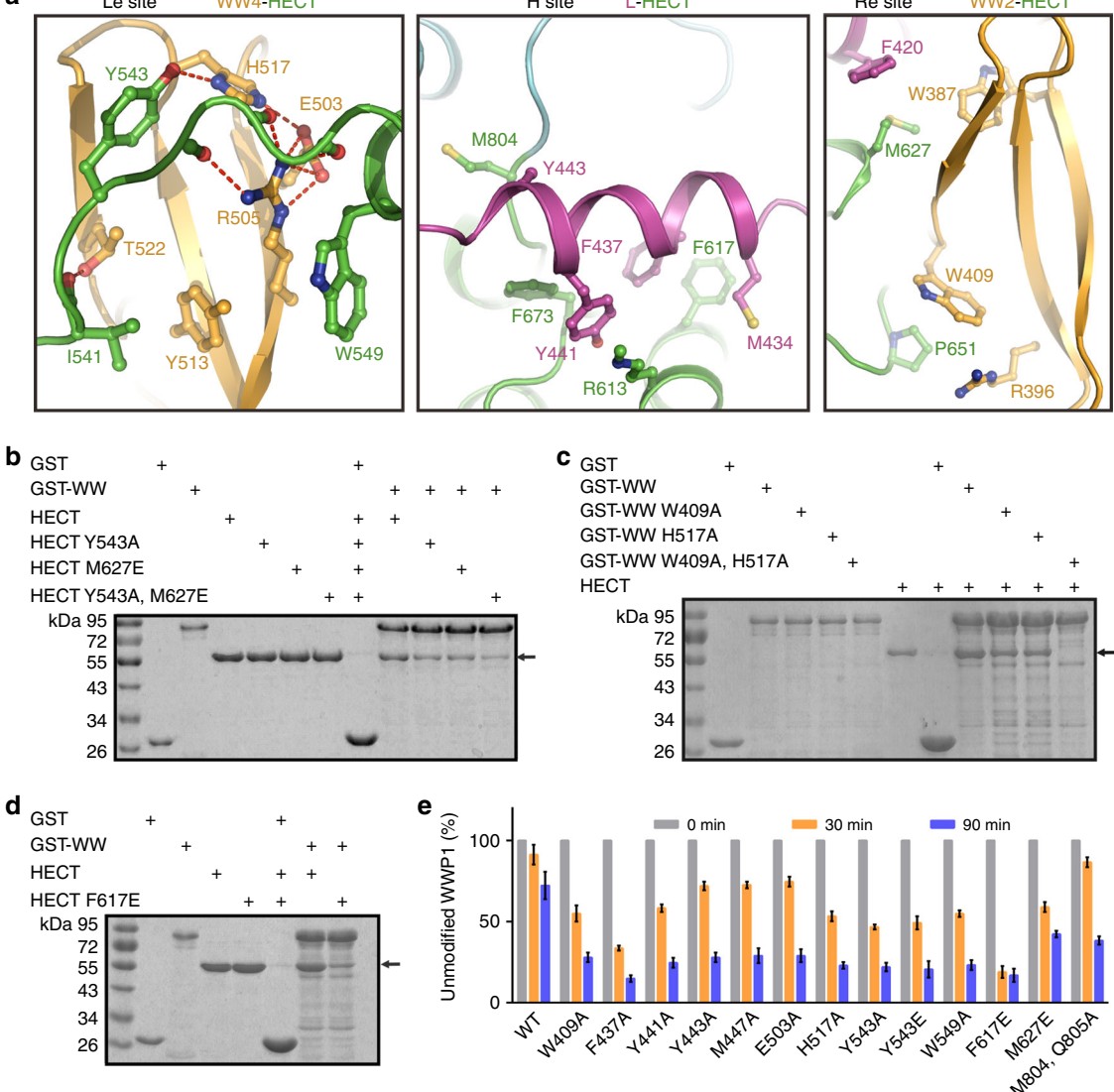

**Fig. 3** WW2L34-HECT interface. **a** The detailed structures of the WW2-HECT, L-HECT, and WW4-HECT interfaces from 2L34HECT and L34HECT. **b**–**d** GST pull-down assay of GST-tagged WWP1 WW12L34 (referred to as WW) with Trx-HECT. The mutations were predicted to impair interactions involving the "Re" site in WW2-HECT (M627E[HECT] and W409A[WW2]), the "Le" site in WW4-HECT (Y543A[HECT] and H517A[WW4]) or the "H" site in L-HECT (F617E[HECT]). **e** Statistical results of the autoubiquitination assay of Trx-tagged WT WWP1 12L34HECT and mutants in Supplementary Fig. 2. The mutations were located in the WW2L34-HECT-binding interface. Data are presented as the mean ± SD of triplicate experiments. Source data are provided as a Source Data file

compared with that observed for the mock group, indicating that the cancer-related migration and proliferation effects of WWP1 are tightly coupled to its ligase activity.

We then tried to identify the potential targets of WWP1 that promoted cell migration. ΔNp63α is the predominant isoform of p63 expressed in epithelial cells, and inhibition of ΔNp63α expression results in elevated cell motility and tumor metastasis[35–37]. Moreover, the fact that ΔNp63α could be ubiquitinated by WWP1 and further degraded through the proteasomal pathway led us to explore whether the constitutively active WWP1 mutants enhanced cell migration through the p63 pathway[38,39]. We first confirmed that, in MCF-10A cells, overexpression of WT WWP1 or its mutants with elevated enzymatic activity indeed promoted the turnover of endogenous ΔNp63α (Fig. 4e). Consistently, the in vitro ubiquitination assay indicated that more ΔNp63α proteins were ubiquitinated by the

constitutively active WWP1 mutants than WT WWP1 (Fig. 4f). Furthermore, restoration of the ΔNp63α level in MCF-10A cells overexpressing WT WWP1 or its mutant partially rescued the cell migration defects (Fig. 4g, h, and Supplementary Fig. 3c), demonstrating the key role of WWP1-mediated ΔNp63α turnover in promoting cell motility. Overall, these results indicated that WWP1 may influence cancer progression by promoting cell proliferation and migration via the regulation of ΔNp63α turnover.

**The multi-lock autoinhibition mode exists in WWP2 and Itch.** The domain organization and primary sequences of WWP2 and Itch are highly similar to those of WWP1 (Supplementary Fig. 4)[32], and these three proteins are thought to be more similar to each other than other Nedd4 family members. The solved crystal

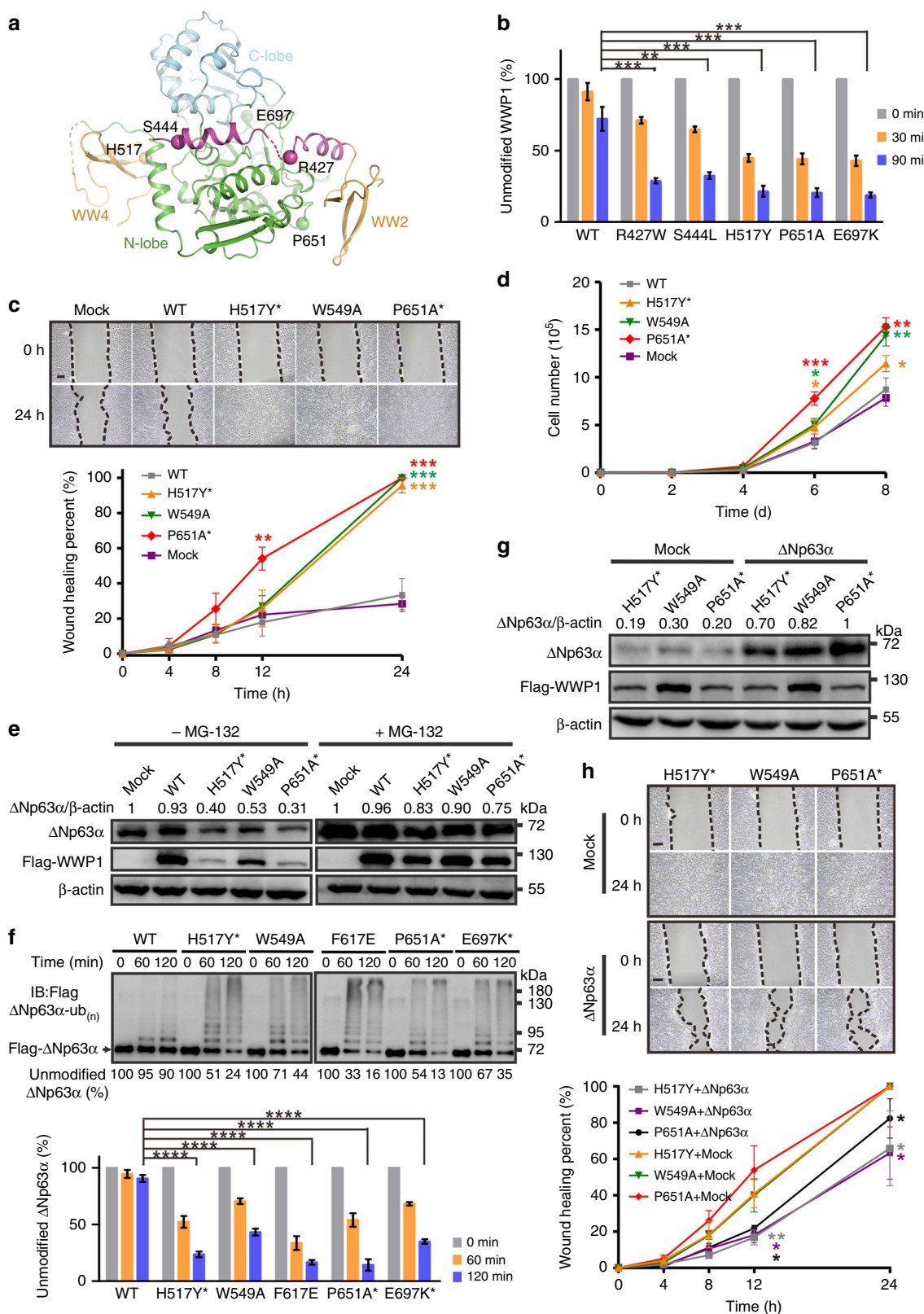

structures of WWP2 2LHECT (PDB ID: 5TJ7 [https://doi.org/10.2210/pdb5TJ7/pdb]), WWP1 2L34HECT (this study), and Itch 12L34HECT (PDB ID: 5XMC [https://doi.org/10.2210/pdb5XMC/pdb]) suggest that L and WW2 bind intramolecularly to the HECT domain and thus regulate the ligase activity of

all three proteins (Supplementary Fig. 1c)[31,32]. Importantly, key residues involved in the WW2L-HECT interaction (e.g., Met627 in WWP1) are highly conserved in all three proteins (Fig. 3a and Supplementary Fig. 4). We noticed that Tyr543, the key residue responsible for binding to WW4 in WWP1, is conserved in

**Fig. 4** Cancer-related mutations elevated autoubiquitination of WWP1 and promoted cell migration by downregulating ΔNp63α in an activity-dependent manner. **a** Structural illustration of cancer-related mutations in WWP1 2L34HECT investigated in this study. **b** Statistical results of the autoubiquitination assay of WT WWP1 and cancer-related mutants in Supplementary Fig. 2. **c–d** Wound-healing assay **c** and cell proliferation assay **d** conducted using MCF-10A cells stably expressing a control vector (Mock), WT WWP1 or mutants (W549A^HECT at the "Le" site and cancer-related H517Y^WW4 and P651A^WW2 at the "Le" site and the "Re" site, respectively). **e** Western blotting analyses of ΔNp63α levels in MCF-10A cells stably expressing WT WWP1 or mutants with or without MG-132 treatment. The weight markers belong to all separate gels. **f** In vitro ubiquitination of ΔNp63α by WT WWP1 or mutants. The weight markers belong to all separate gels. **g** ΔNp63α or a control vector (Mock) were overexpressed in MCF-10A cells stably expressing various WWP1 mutants, and the cells were then subjected to western blotting analyses of the ΔNp63α level. **h** ΔNp63α or a control vector (Mock) were overexpressed in MCF-10A cells stably expressing various WWP1 mutants, and the cells were then subjected to a wound-healing assay. Scale bars, 200 μm. Cancer-related mutants are marked with asterisks. Data are presented as the mean ± SD of triplicate experiments; ns, not significant, *p < 0.05, **p < 0.01, ***p < 0.001, and ****p < 0.0001 based on one-way analysis of variance (ANOVA) with Tukey's multiple comparison test. Source data are provided as a Source Data file

WWP2 and Itch (Supplementary Fig. 4), which suggests the possibility that the multi-lock autoinhibition mode of WWP1 may be conserved in WWP2 and Itch as well. We tested this hypothesis in several ways. We first confirmed that compared with the inactive 2L34HECT, deletion of either WW34 (2LHECT) or WW2 (L34HECT) led to significant elevation of WWP2 ligase activity, whereas deletion of either L (234HECT) or WW234 (L-HECT) led to marked autoubiquitination of WWP2 (Fig. 5a, b); this confirmed the critical role of L, WW2, and WW34 in enzyme inhibition.

The GST pull-down experiments further validated this assumption. Although WW2L and LWW34 could interact with HECT, the entire WW region exhibited the strongest binding. In sharp contrast, WW1, WW2, WW34, or L alone could not form a stable complex with HECT (Supplementary Fig. 5a). Moreover, when WW in WWP2 robustly interacted with HECT (WT), the Y491A^HECT (corresponding to Tyr543 in WWP1) and M575E^HECT (corresponding to Met627 in WWP1) mutations, which were predicted to impair packing in WW4-HECT and WW2-HECT, respectively, significantly weakened the interaction between WW and HECT in WWP2, and the Y491A, M575E double mutation completely eliminated the interaction (Fig. 5c). Accordingly, the W358A^WW2 (corresponding to Trp409 in WWP1) and H465A^WW4 (corresponding to His517 in WWP1) mutations, which were predicted to impair the packing of WW2-HECT and WW4-HECT, respectively, significantly weakened the interaction between WW and HECT in WWP2, and the W358, H465A double mutation completely disrupted the interaction (Fig. 5d). Finally, both the Y491A and M575E mutations in WWP2 12L34HECT resulted in elevated ligase activity compared to that of the WT (Fig. 5e), possibly due to the impaired intramolecular interaction.

Similar results were observed in Itch. Both L34HECT and 12LHECT possessed elevated ligase activities compared with the inactive 12L34HECT (Fig. 5f, g). Though WW12L could robustly interact with HECT in Itch, isolated WW12 or L could not[32]. Mutations that were predicted to impair interactions with the "Re" site in WW2-HECT (W347A^WW2 and M569E^HECT) or the "Le" site in WW4-HECT (Y485A^HECT and H460A^WW4) weakened the WW–HECT interaction of Itch, and the double mutation of both sites (Y485A, M569E or W347, H460A) completely eliminated the interaction (Fig. 5h, i). Moreover, the M569E and Y485A mutations in Itch (12L34HECT) resulted in elevated ubiquitination activity (Fig. 5j).

We noticed that the WW4-HECT interaction was not observed in the Itch structure (PDB:5XMC) despite the fact that WW3 and WW4 were present in the construct[32]. Structural analysis revealed that although the WW4-binding "IAY" motif from the N-terminal extension of WWP1 HECT is completely conserved in Itch, there are still some variations in the WW4-HECT packing surface. For example, Trp549^HECT which interacts with WW4 in

both WWP1 2L34HECT and L34HECT (Fig. 3a and Supplementary 1g), is substituted with Ala in Itch, that may weaken the WW4-HECT interaction. Moreover, when we superimpose WWP1 2L34HECT structure to Itch structure (PDB:5XMC), although WW2L-HECT in both structures fit very well (Supplementary Fig. 1c), the "Le" site in Itch is occupied by the WW2 domain of a symmetric molecule (Supplementary Fig. 5b). Such competition between a symmetric Itch' WW2 and WW4 toward HECT during crystallization may be a reason why the WW4-HECT interaction was not observed in the crystal structure of Itch 12L34HECT.

Overall, the above biochemical analysis indicated that WWP2 and Itch might adopt a similar multi-lock mode as WWP1 to keep the enzymes in an inactive state.

**Nedd4/4 L and Smurf2 adopt a varied multi-lock autoinhibition.** As a characteristic structural component of Nedd4 family E3 ligases, multiple WW domains also exist in other Nedd4 family members, including Nedd4/4 L and Smurf1/2, and the WW domains in those E3s were shown to contribute to ligase regulation (Supplementary Fig. 6a). Distinct from WWP1/2 and Itch, in which ligase regulation is driven mainly by the WW region, the C2 domains in Nedd4/4 L and Smurf2 (and likely Smurf1 as well) play a key regulatory role by occupying the "Re" site in the HECT domain[26–28]. Consistent with previous studies, removal of C2 from Nedd4/4 L (1L234HECT), Smurf1, and Smurf2 (1L23HECT) dramatically increased their enzymatic activity (Fig. 6a–d and Supplementary Fig. 6a–e). Owing to steric hindrance, the WW domain(s) in Nedd4/4 L and Smurf1/2 are unlikely to interact with the HECT domain through the same "Re" site. Interestingly, the key residues in WWP1 involved in binding to WW4 (e.g., Y543) and L (e.g., F617) are completely conserved in other Nedd4 family E3 HECT domains (Supplementary Fig. 4), implying that the "Le" and "H" sites may exist in Nedd4/4 L and Smurf1/2 and participate in ligase regulation. However, the WW2L module observed in Itch and WWP1/2 does not exist in Nedd4/4 L and Smurf1/2.

We then searched for regulatory element(s) by introducing domain truncations. Deletion of WW1 (but not the other WW domains) or the L that followed it significantly increased the ligase activity of Nedd4/4 L and Smurf2, and a combinatorial deletion of both WW1 and L resulted in comparable activity with HECT (Fig. 6a–d and Supplementary Fig. 6a–e), implying that WW1 in those E3s may act as the "left ear" by coupling with L to sequester HECT at the "Le" site. Unfortunately, our extensive attempts to generate crystals of Nedd4/4 L or Smurf1/2 in their fully inactive or partially active states all failed. Sequence alignment of L from Nedd4/4 L and Smurf2 revealed a certain level of conservation, although Smurf2 has a much longer insertion (~30 aa) between WW1 and L when compared with

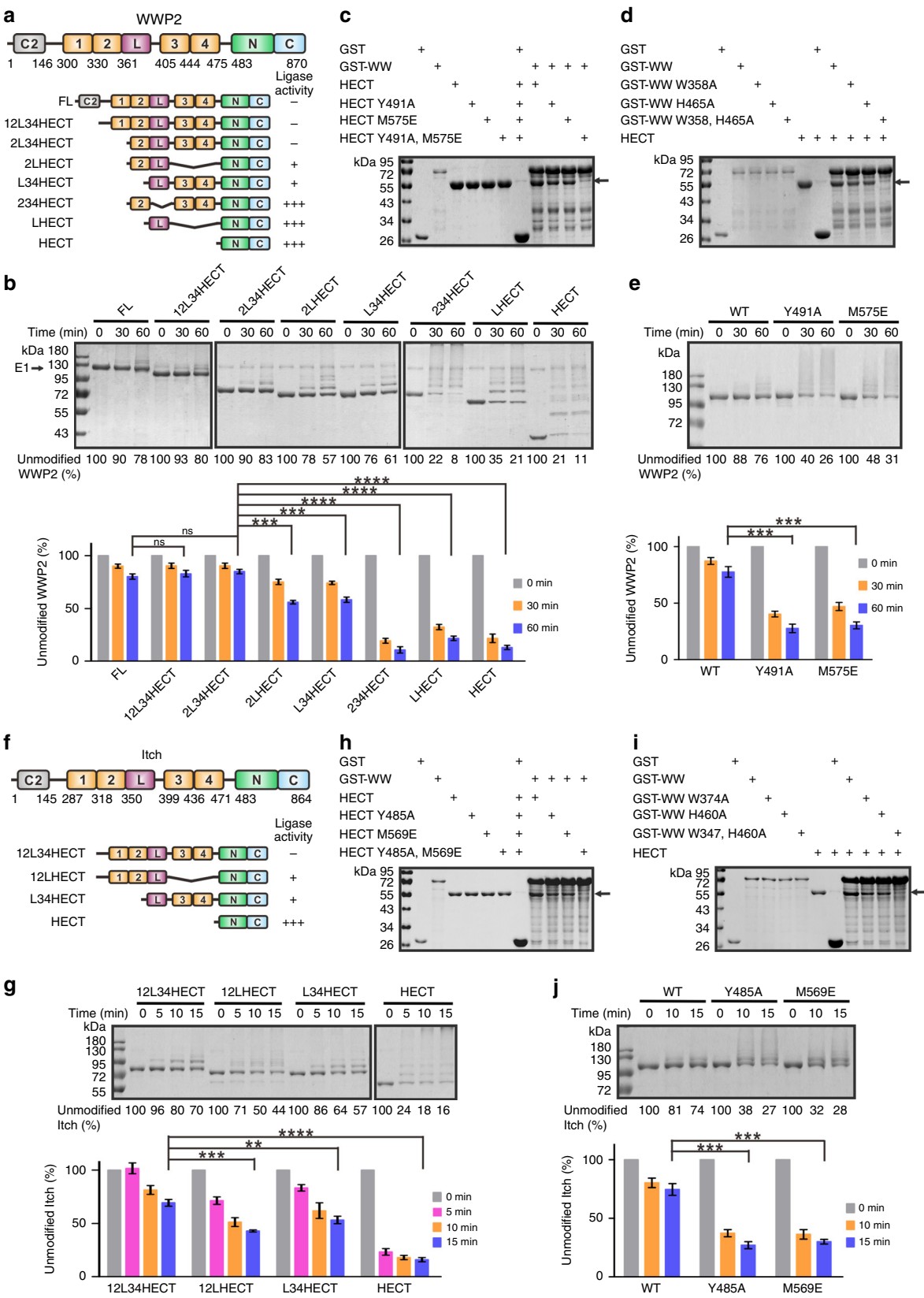

Nedd4/4 L (Fig. 6e). The GST pull-down result further demonstrated that this L (together with WW1) was sufficient for HECT binding in Nedd4 (Fig. 6f). We then built a structural model of Nedd4 1LHECT using WWP1 L34HECT as the template (Fig. 6g).

In this model, WW1 and L are anchored to the "Le" and "H" sites, respectively, though the orientation of L in Nedd4 is reversed compared with that in WWP1/2 and Itch. Specifically, the conserved Tyr518[HECT] inserts into the canonical "NPxY"

**Fig. 5** WWP2 and Itch might adopt the same multi-lock regulation mechanism as WWP1. **a** Schematic of WWP2 domains showing an enzymatic activity summary derived from the autoubiquitination assay in **b**. **b** Autoubiquitination assay of Trx-tagged full-length (FL) WWP2 and various fragments. The weight markers belong to all separate gels. **c–d** GST pull-down assay of WWP2 GST-WW with Trx-HECT. Mutations that were predicted to impair interactions at the "Re" site in WW2-HECT ($W358A^{WW2}$ and $M575E^{HECT}$) or the "Le" site in WW4-HECT ($Y491A^{HECT}$ and $H465A^{WW4}$) weakened the WW–HECT interaction of WWP2, and the double mutation of both sites ($Y491A^{HECT}$, $M575E^{HECT}$, and $W358A^{WW2}$, $H465A^{WW4}$) completely disrupted the interaction. **e** The in vitro autoubiquitination assay of WWP2 12L34HECT $Y491A^{HECT}$ and $M575E^{HECT}$ mutants. **f** Schematic of Itch domains showing a summary of enzymatic activity derived from the autoubiquitination assay in **g**. **g** Autoubiquitination assay of various Trx-tagged Itch fragments. The weight markers belong to all separate gels. **h–j** GST pull-down assay of Itch GST-WW with Trx-HECT. Mutations that were predicted to impair interactions at the "Re" site in WW2-HECT ($W347A^{WW2}$ and $M569E^{HECT}$) or the "Le" site in WW4-HECT ($Y485A^{HECT}$ and $H460A^{WW4}$) weakened the WW–HECT interaction with Itch, and double mutation of both sites ($Y485A^{HECT}$, $M569E^{HECT}$, and $W347^{WW2}$, $H460A^{WW4}$) completely disrupted the interaction. **j** In vitro autoubiquitination assay of Itch 12L34HECT $Y485A^{HECT}$ and $M569E^{HECT}$ mutants. For all ubiquitination assays, the statistics showing enzymatic activity are shown below. Data are presented as the mean ± SD of triplicate experiments; ns, not significant, $*p < 0.05$, $**p < 0.01$, $***p < 0.001$, and $****p < 0.0001$ based on one-way analysis of variance (ANOVA) with Tukey's multiple comparison test. Source data are provided as a Source Data file

motif-binding site in WW1 to form a hydrogen bond with $His212^{WW1}$ (Fig. 6g). $R247^L$ forms a salt bridge with $E591^{HECT}$, whereas $F244^L$ packs with $F594^{HECT}$ through hydrophobic interaction. Note that all of the above residues are highly conserved among Nedd4/4L and Smurf2 (Fig. 6e and Supplementary Fig. 4). In line with the structural model, point mutations at the modeled WW1-HECT "Le" interface ($H212A^{WW1}$ and $Y518A^{HECT}$ in Nedd4 and $H178A^{WW1}$ and $Y368A^{HECT}$ in Smurf2) disrupted the interaction between the entire WW domain region and HECT, whereas mutations in other WW domains, such as the triple mutation $H369^{WW2}$, $H442^{WW3}$, $H494^{WW4}$A (hereafter referred to as H369,442,494 A) in Nedd4 and the double mutation $H271^{WW2}$, $H318^{WW3}$A (hereafter, referred to as H271,H318A) in Smurf2 barely weakened the WW–HECT interaction (Fig. 6h, i). In addition, point mutations at the predicted L-HECT "H" interface ($F244A^L$, $R247A^L$, $E591A^{HECT}$, $F594E^{HECT}$ in Nedd4 and $Y241A^L$, $R244A^L$, or $E440A^{HECT}$, $Y443E^{HECT}$ in Smurf2) completely disrupted the WW–HECT interaction (Fig. 6j, k). Moreover, the above mutations led to elevated ligase activity compared to the WT enzymes (Fig. 6l, m), which further supported our structural model. In summary, Nedd4/4 L and Smurf2 might utilize a varied multi-lock regulation mechanism in which C2, WW1, and L pack with the conserved "Re", "Le", and "H" sites in HECT, respectively, to keep the enzymes in their autoinhibited state.

## Discussion

The most important finding of this study is that every one of the WWP1/2, Itch, Nedd4/4 L, and Smurf2 E3 ligases employs a multi-lock mechanism to keep it in an autoinhibited state. In addition to the previously reported "Re" and "H" sites, which are responsible for WW2 (WWP2 and Itch) or C2 (Smurf2 and Nedd4/4 L) and L (WWP2 and Itch) binding, respectively[26–28,31,32], a new "Le" site was discovered here. Through combinatorial deployments of either the "Re"/"H" or "Le"/"H" sites, the Nedd4 E3s can be kept in a partially active state. When the "Re", "H", and "Le" sites are all engaged (i.e., when the multi-locks act simultaneously), the Nedd4 E3s are in the fully autoinhibited states. Such multilayered regulatory mechanisms presumably can produce gradual activation of the enzymes involved in substrate ubiquitination and turnover of diverse cellular functions. Spatiotemporal post-translational modifications and/or activator binding at these regulatory sites could induce these E3s into partially or fully active states with finely tuned ligase activity. However, pathological mutations at these sites may impair such regulatory mechanisms and thus cause misregulated enzyme activity in E3s (Fig. 4), leading to numerous diseases such as cancers.

How does the spatiotemporal activation of Nedd4 E3s occur physiologically? Although it has been suggested that tyrosine

phosphorylation within L could disrupt the autoinhibition of WWP2[31], how this occurs remains unclear, as tyrosine kinases normally phosphorylate protein fragments with extended conformations[40]. Our data suggested that the WW2 or WW34 domains might stabilize the two ends of helical L during interaction with HECT (Figs. 1 and 2). Thus, phosphorylation of L might only occur when some factor(s) (e.g., PY motif-containing substrates/adaptors) engage the neighboring "Re" or "Le" site to dissociate the WW domain(s) from the HECT domain and release the L segment from the HECT domain for kinase recognition. Dissociation of the WW domains and tyrosine phosphorylation on L may cooperate with each other to induce a fully active state. Moreover, we noticed that the conserved Tyr543 on the "Le" site of WWP1 is within a flexible region and is suitable for phosphorylation by tyrosine kinases. According to the PhosphoSitePlus database[41] (https://www.phosphosite.org), Tyr543 of WWP1 is phosphorylated in human T-cell leukemia Jurkat cells. Importantly, introduction of a negative charge at Tyr543 in the form of a Y543E mutation led to elevated ligase activity compared to the WT enzymes (Fig. 3e), suggesting the potential role of Tyr543 phosphorylation in the progression of acute myeloid leukemia[19], though glutamate and phosphotyrosine are structurally different. Considering the high conservation of Tyr543, its phosphorylation might play a common regulatory role in Nedd4 family members.

Based on their autoinhibitory mechanisms, Nedd4 E3s may be grouped into two subfamilies: one comprised of WWP1/2 and Itch, in which the versatile "Re" site in HECT is occupied by WW2, and another that is comprised of Nedd4/4 L and Smurf2, in which the same site is occupied by C2; in both subfamilies, the conserved "Le" and "H" sites in HECT are occupied by the WW domain (WW4 in WWP1/2 and Itch and WW1 in Nedd4/4 L and Smurf2) and L, respectively. Presumably, the different regulatory modes determine their functional specificity (Fig. 7). For WWP1, WWP2 or Itch, the WW region is the main element involved in autoinhibition. These three E3s could be activated when interacting with multiple PY-containing adaptors (such as Ndfip1) or substrates and thus promote temporally specific targeted ubiquitination, such as that resulting from the Itch-mediated temporal degradation of JunB after T-cell activation, which promotes Ndfip1 expression[42]. However, for other Nedd4 E3s, neither the weak interaction between $Ca^{2+}$/phospholipids and C2[27,28] nor the binding of PY motif-containing targets to the WW domains can fully activate E3. Only when the membrane targeting of the C2 domain and the target binding of the WW domain are simultaneously achieved will the ligase activity of a Nedd4 E3 be fully induced; this provides enhanced, specific spatial control of E3 activity. Finally, different combinations of C2 domain-mediated membrane localization and WW domain-mediated target binding

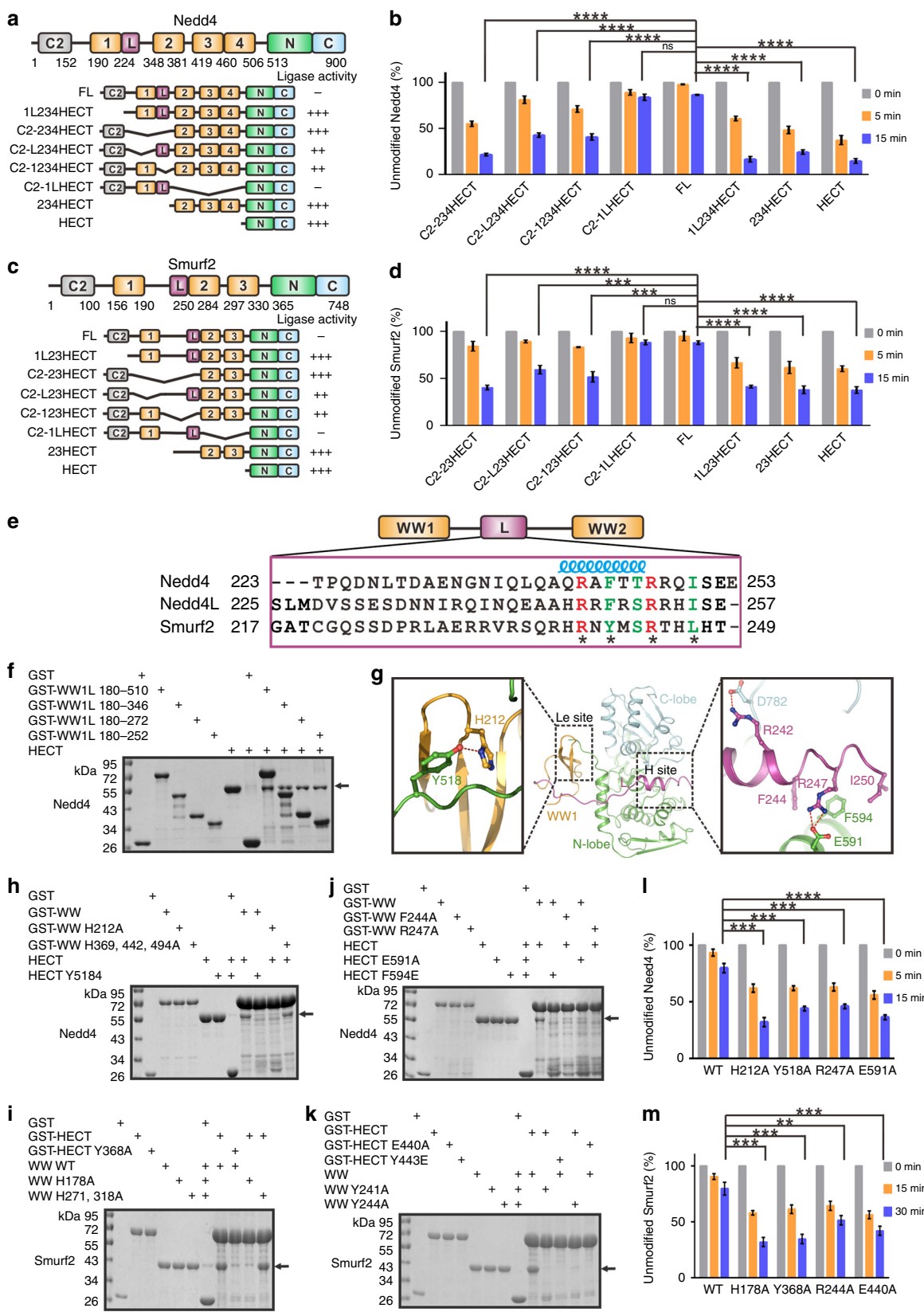

may allow for another level of fine-tuning during the regulation of the activity s of different Nedd4 members.

Considering the important biological functions of HECT family E3s, the above-described regulatory sites could be potential target sites for manipulating ligase activity. Zhang et al.[43] designed multiple ubiquitin variants that targeted the "Re" site to promote Nedd4 family E3 ligase activity. The newly identified "Le" site in WWP1 might also have diverse regulatory functions.

**Fig. 6** C2 and WW1L lock Nedd4 and Smurf2 in an inactive state. **a** Schematic of Nedd4 domains showing a summary of enzymatic activity derived from the autoubiquitination assay in **b**. **b** Statistical results of the autoubiquitination assay of various Trx-tagged Nedd4 fragments in Supplementary Fig. 6b. **c** Schematic of Smurf2 domains showing a summary of enzymatic activity derived from the autoubiquitination assay in **d**. **d** Statistical results of the autoubiquitination assay of various GST-tagged Smurf2 fragments in Supplementary Fig. 6e. **e** The primary sequence alignment of L in Nedd4, Nedd4L, and Smurf2. The identical residues are colored in red, and the highly conserved residues are colored in green. The residues involved in packing with HECT are marked with asterisks. **f** GST pull-down assay of GST-tagged Nedd4 WW1L and various truncated fragments with HECT. **g** Model of Nedd4 1LHECT. **h**–**k** GST pull-down assay of Nedd4 GST-WW with Trx-HECT **h** and **j** and Smurf2 GST-HECT with Trx-WW **i** and **k.** For Nedd4, mutations at the predicted WW1-HECT "Le" site interface (H212A$^{WW1}$ and Y518A$^{HECT}$) **h** and the L-HECT "H" site interface (F244A$^L$, R237A$^L$, E591A$^{HECT}$, and F594E$^{HECT}$) **j** disrupted the interaction between the entire WW region and HECT. For Smurf2, mutations at the predicted WW1-HECT "Le" site interface (H178A$^{WW1}$ and Y368A$^{HECT}$) **i** and the L-HECT "H" site interface (Y241A$^L$, R244A$^L$, E440A$^{HECT}$, and Y443E$^{HECT}$) **k** disrupted the interaction between the entire WW domain region and HECT. **l** The in vitro autoubiquitination assay of Nedd4 FL mutants (H212A$^{WW1}$, Y518A$^{HECT}$, R247A$^L$, or E591A$^{HECT}$). **m** The in vitro autoubiquitination assay of Smurf2 FL mutants (H178A$^{WW1}$, Y368A$^{HECT}$, R244A$^L$, or E440A$^{HECT}$). Data are presented as the mean ± SD of triplicate experiments; ns, not significant, *$p < 0.05$, **$p < 0.01$, ***$p < 0.001$, and ****$p < 0.0001$ based on one-way analysis of variance (ANOVA) with Tukey's multiple comparison test. Source data are provided as a Source Data file

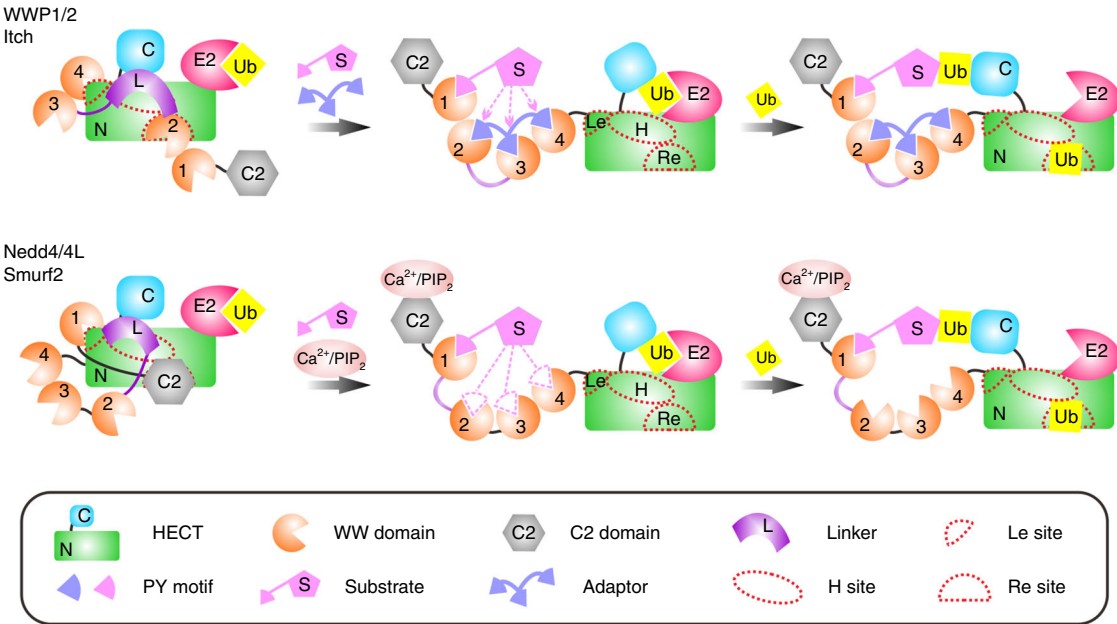

**Fig. 7** Model of the regulation of Nedd4 family E3s. A versatile multi-lock regulatory mechanism is utilized for the fine-tuning of Nedd4 family E3 ligase activity. Three conserved regulatory sites ("Re", "Le", and "H") have been identified in the HECT domain. Although combinatorial regulation of either "Re"/ "H" or "Le"/"H" sites could suppress the ligase activity of Nedd4 E3s to a certain extent, only the multi-lock mode involving all three sites can keep Nedd4 E3s in the complete inactive state. For Itch and WWP1/2, E3s could be activated via multivalent PY-WW interactions with adaptors (such as Ndfip1) or substrates and thus promote temporally specific target ubiquitination. For other Nedd4 E3s, the C2 domain is also involved in autoinhibition. The enzymes undergo spatially specific activation upon binding to their respective target proteins (via WW domains) in restricted cellular regions (via the C2 domain)

In this study, we showed that Tyr543$^{WWP1}$ occupied the canonical PY motif-binding site of WW4 and also locked WWP1 in its inactive state (Fig. 3). A recent study reported that ubiquitination of Rsp5 at Lys432 (corresponding to the conserved Lys550$^{WWP1}$ on the "Le" site) could suppress its activity by inducing its oligomerization (Supplementary Fig. 7)[24]. Similarly, the N-terminal extension of the HECT domain (which partially overlaps with the "Le" site) from non-Nedd4 E3 HUWE1 mediates its dimerization and thus maintains low activity in E3[25], and the conserved Tyr3988$^{HUWE1}$ (corresponding to Tyr543$^{WWP1}$) provides a key contribution to dimer formation through hydrophobic interactions (Supplementary Fig. 7). These findings suggest that the "Le" site may act as a versatile regulatory site in HECT domains.

WWP1 has been frequently found to be upregulated or mutated in human tumors and cancers, although the underlying mechanism remains unclear[17–19]. Within the COSMIC cancer somatic mutation database[34] (http://cancer.sanger.ac.uk/cosmic), 85 out of 159 mutations occur in the WW2L34 and HECT domains (Supplementary Fig. 3a). The crystal structure of WWP1 2L34HECT presented in this study provides a useful tool that can be used for interpreting the effects of numerous mutations (48 out of 85) found in this region in patients with cancers and other diseases. Importantly, all cancer-related mutations of WWP1 within the WW–HECT packing interface led to elevated enzymatic activity, thus promoting cell migration and proliferation possibly by mediating ΔNp63α turnover. Presumably, mutations important for protein folding would disrupt the overall structure and thus the substrate ubiquitination of WWP1. We propose that WWP1 mutations found in patients within the 2L34HECT region that fall into the above two categories will have a higher chance of being relevant to disease and are thus valuable for clinical

diagnosis and investigation. Note that the substrate ubiquitination assays conducted in vitro and in MCF-10A cells both showed that ΔNp63α could barely be ubiquitinated by WT WWP1, even though E3 was overexpressed (Fig. 4e, f), which implies that WWP1 is still kept in an inactive state even when the protein level has been elevated. However, its dysregulated activation though effectors or mutations will lead to the uncontrolled turnover of its targets (e.g., tumor suppressors), thus resulting in tumor progression and metastasis.

## Methods

**Protein expression and purification**. Various fragments from human WWP1/2, Nedd4/4 L, and Smurf1/2, and mouse Itch (Supplementary Table 1) were individually cloned into a modified version of pGEX-6P-1 or pET-32a vectors, with the resulting protein containing a GST-His$_6$ or Trx-His$_6$ tag followed with a Prescission protease-cutting site in its N terminus. All of the mutations used in this study were created through standard PCR-based mutagenesis method and confirmed by DNA sequencing. All primer sequences used in this study were listed in Supplementary Table 2. Recombinant proteins were expressed in *Escherichia coli* BL21 (DE3) host cells (New England Biolabs). After induction at 16 °C for 16 h, the cells were harvested and resuspended in the buffer containing 50 mM Tris (pH 8.0), 500 mM NaCl, 10 mM imidazole, and 1 mM phenylmethylsulfonyl fluoride. The cells were then lysed by sonication and the cell debris was removed by centrifugation. The supernatant was incubated with Ni$^{2+}$-NTA agarose and the bound proteins were eluted by the binding buffer supplied with 250 mM imidazole. The eluted proteins were purified by size-exclusion chromatography (SEC) equilibrated with the buffer containing 50 mM Tris (pH 8.0), 100 mM NaCl, 1 mM DTT and 1 mM EDTA. Unless otherwise specified, GST-His$_6$ tagged proteins were used for pull-down assays and all other studies were performed with Trx-His$_6$ tagged proteins. For crystallization, the N-terminal Trx-tagged fragments of recombinant proteins were cleaved by digesting fusion proteins with Prescission protease (50 μg protein with 1 μl protease, Sigma, GE27-0843-01) at 4 °C, and the proteins were purified by another step of SEC.

Human ΔNp63α was sub-cloned into pCMV-Tag2b, and transfected to HEK293T cells (from ATCC) using polyethylenimine transfection reagent (Polysciences). After cultured at 37 °C for 72 h, cells were harvested and lysed in lysis buffer (50 mM Tris pH 7.4, 150 mM sodium chloride, 1% Nonidet P-40, 10 mM sodium fluoride, 1 mM sodium metavanadate, 1 mM phenylmethylsulfonyl fluoride and protease inhibitors) at 4 °C for 30 min. The lysates were clarified by centrifugation at 21,130 × *g* at 4 °C for 30 min. Supernatants were mixed with anti-Flag M2 affinity gel (Sigma, A2220) and incubated at 4 °C for 2 h. After extensive wash with the lysis buffer, the target protein captured by affinity beads was eluted with commercially synthesized Flag peptide (150 ng/μl) and used for ubiquitination assay.

**In vitro ubiquitination assay**. For the autoubiquitination assay, 800 nM Nedd4 family E3s or their truncated forms were incubated with 60 nM E1 (UBE1), 400 nM E2 (UBCH5A), and 100 μM HA-ubiquitin in ubiquitination reaction buffer containing 50 mM Tris (pH 7.5), 5 mM MgCl$_2$, 1 mM DTT, and 5 mM ATP.

For substrate ubiquitination assay, the purified Flag-ΔNp63α were incubated with 60 nM E1 (UBE1), 400 nM E2 (UBCH5A), 800 nM WT E3s or mutants, and 100 μM HA-ubiquitin in ubiquitylation reaction buffer.

The reactions were initiated by adding ATP and carried out at 37 °C. The reactions were quenched by mixing the reaction mixture with SDS loading dye at indicated time points. Then samples were resolved by sodium dodecyl sulfate polyacrylamide gel electrophoresis (SDS-PAGE), stained with Coomassie brilliant blue (CBB) or used for immunoblotting. The unmodified E3s or substrate bands at distinct time points shown in the figures were quantified and normalized to the zero time points. All assays were repeated at least three times showing similar results.

**E2–E3 transthiolation assay**. E2–E3 transthiolation assay was conducted according to previous method[27,44]. In brief, 800 nM Trx-His-tagged E3s were truncated by five residues from the C terminus (ΔCT) to inhibit their auto-ubiquitination and incubated with 60 nM E1 (UBE1), 400 nM E2 (UBCH5A), and 100 μM ubiquitin at 37 °C in ubiquitylation reaction buffer containing 50 mM Tris (pH 7.5), 5 mM MgCl$_2$, 0.1 μM DTT and 5 mM ATP for 5 or 10 min. The reactions were stopped with SDS-PAGE loading buffer without (top panel) or with (bottom panel) 100 mM DTT (bottom panel). E3s were detected by staining with CBB.

**Immunoblotting**. Proteins were boiled in SDS-PAGE loading buffer and subjected to SDS-PAGE. The proteins were transferred to a 0.45-μm nitrocellulose membrane (Millipore, IPVH00010), and the nitrocellulose membrane was blocked with 3% bovine serum albumin in tris-buffered saline with Tween 20 (TBST, 20 mM Tris-HCl, pH 7.4, 137 mM NaCl and 0.1% Tween 20) buffer at room temperature for 1 h, followed by incubation with anti-Flag (ABclonal, AE005, 1:3000), or anti-p63 (CST, 13109 S, 1:1000) antibody at 4 °C overnight. The membranes were washed three times with TBST buffer, incubated with horseradish peroxidase-conjugated goat anti-mouse antibody (ABclonal, AS003, 1:4000) or anti-rabbit antibody (ABclonal, AS014, 1:4000) and visualized on a LAS4000 chemiluminescent imaging system (GE HealthCare).

**Crystallography**. Freshly purified WWP1 2L34HECT was concentrated to 7 mg/ml in Buffer A containing 50 mM Tris (pH 8.0), 500 mM NaCl, 1 mM ethylenediaminetetraacetic acid (EDTA) and 1 mM DTT. Crystals of 2L34HECT were grown by the hanging-drop vapor diffusion method at 16 °C in a reservoir solution containing 100 mM Tris-HCl pH 7.0 and 15% v/v reagent alcohol, and then were soaked in crystallization solution containing 30% glycerol for cryoprotection. Freshly purified WWP1 L34HECT was concentrated to 15 mg/ml in Buffer A for crystallization. Crystals of L34HECT were grown by the hanging-drop vapor diffusion method at 16 °C in a reservoir solution containing 0.1 M Sodium malonate pH 5.0, 12% w/v Polyethyleneglycol 3350, and then were soaked in crystallization solution containing 25% glycerol for cryoprotection. The diffraction data of the crystals were collected at the Shanghai Synchrotron Radiation Facility (SSRF) beamline BL17U1 in China at a wavelength of 0.9792 Å. The data were processed and scaled with HKL2000[45]. The phase problem of the 2L34HECT or L34HECT structure was solved by molecular replacement using WWP1 HECT structure (PDB ID: 1ND7 [https://doi.org/10.2210/pdb1ND7/pdb])[21] as the search model against the 2.3 Å and 2.5 Å resolution data sets, respectively. The initial model was further rebuilt, adjusted manually with Coot[46] and refined by the phenix.refine program of PHENIX[47]. The final models had 97.2% (2L34HECT) or 95.3% (L34HECT) of the residues in the favored region of the Ramachandran plot with no outliers, respectively. The final refinement statistics are summarized in Table 1.

**GST pull-down assay**. GST or GST fusion proteins were first loaded onto 40 μl GSH-Sepharose 4B slurry beads and then incubated with potential binding partners in an assay buffer containing 50 mM Tris (pH 8.0), 100 mM NaCl, 1 mM DTT, and 1 mM EDTA at 4 °C for 1 h. After washing three times, proteins captured by affinity beads were eluted by boiling, resolved by 12% SDS-PAGE, and detected by CBB.

**Cell culture and virus infection**. MCF-10A cells (from ATCC) were maintained in 1:1 mixture of DMEM (Hyclone) and Ham's F-12 medium (Invitrogen, 11330032), supplemented with 20 ng/ml epidermal growth factor (PeproTech AF-100-15), 100 ng/ml cholera toxin (Sigma, C-8052), 10 μg/ml insulin (Sigma, I-1882), 500 ng/ml hydrocortisone (Sigma, H-0888), and 5% horse serum (Invitrogen, 16050122).

Retroviruses were employed to stably express WWP1 using pLVX vector. Retroviral vectors were co-transfected into 293 T cells using psPAX2 and pMD2G as packing plasmids. After 36 h, viruses in the supernatant were collected and used to infect MCF-10A cells in the presence of 8 μg/ml polybrene. Stably expressed cells were selected in 1 μg/ml puromycin (Gibco, A1113802) for 1 week.

**Cell transfection**. Cell transfection was used for ΔNp63α ectopic expression in WWP1 stably expressing MCF-10A cells. Human ΔNp63α was sub-cloned into pCMV-Tag2b, and transfected to MCF-10A cells using polyethylenimine transfection reagent (Polysciences, 23966). After cultured at 37 °C for 24 h, the cells were collected and used for wound-healing assay.

**Cell proliferation assay**. MCF-10A cells were seeded at 1000 cells/well into six-well plates in triplicate and cell numbers were counted using Cellometer Mini (Nexcelom Bioscience) per day for 8 days, respectively.

**Wound-healing assay**. MCF-10A cells were plated in each well of Culture Inserts (Ibidi, 80209) and cultured with serum free medium at 37 °C. After 12 h, the Culture Inserts were removed. At indicated time intervals, cells were photographed under a light microscope. The wound-healing area at distinct time points shown in the figures were quantified and normalized to the zero time points. Data are presented as mean ± SD of triplicate experiments.

**Protein structure modeling**. The structural model of Nedd4 1LHECT was generated by COOT using the crystal structure and density of WWP1 L34HECT in this study as a template, guided by the sequence alignment of Nedd4 with WWP1 performed by Clustal Omega (http://www.ebi.ac.uk/Tools/msa/clustalo/), protein secondary structure predicted by The PSIPRED Protein Sequence Analysis Workbench (http://www.bioinf.cs.ucl.ac.uk/psipred/) and the charged/hydrophobic properties of conserved residues in L. The final model was refined by energy minimization using GROMACS[48]. The three-dimensional structure of Nedd4 1LHECT was analyzed by using PyMOL.

**Statistical analysis**. Statistical parameters including the definitions and exact values of *n* (e.g., number of experiments, number of cells, etc.) are reported in the corresponding figure legends. All data are presented as mean ± SD of triplicate experiments; ns, not significant, *$p < 0.05$, **$p < 0.01$, ***$p < 0.001$, and ****$p < 0.0001$ using one-way analysis of variance with Tukey's multiple comparison test.

None of the data were removed from our statistical analysis as outliers. All statistical data were conducted in GraphPad Prism 6. All experiments related to cell cultures and imaging studies were performed in blinded fashion.

**Reporting summary**. Further information on research design is available in the Nature Research Reporting Summary linked to this article.

## Data availability

All the relevant data supporting the findings of this study are available within the article, or from the corresponding author on reasonable request. Coordinates of the crystal structures of WWP1 2L34HECT and L34HECT have been deposited in the Protein Data Bank under the accession code 6J1X and 6J1Y, respectively. The source data underlying Figs. 1–6 and Supplementary Figs. 1, 2, 5, and 6 are provided as a Source Data file.

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

## Acknowledgements

We thank Dr. Mingjie Zhang for valuable discussions; Drs. Hongrui Wang, Ceshi Chen, and Suling Liu for WWP1/2, Nedd4/4L, and Smurf1/2 genes and MCF-10A cells; the staff of beamlines BL17U1 and BL18U1 at SSRF and National Center for Protein Sciences Shanghai for data collection. This work was supported by grants from the National Natural Science Foundation of China (31670730, 31871394, 31422015), the Ministry of Science and Technology of the People's Republic of China (2014CB910201), the Shanghai Municipal Education Commission (14SG06), Shanghai Municipal Science and Technology Major Project (2018SHZDZX01) and ZJLab.

## Author contributions

Z.W., Z.L. and W.W. conceived the research and analyzed data. Z.W., Z.L., X.C., J.L, W.Y., S.H. and A.G. performed the experiments. Z.L. solved the crystal structures and built the structural model. Q.Y.L. helped with the wound-healing assay. Z.W., Y.M. and W.W. wrote the manuscript. W.W. coordinated the research.

## Additional information

**Competing interests:** The authors declare no competing interests.

