## [Peer Review File · Nature Communications]

Reviewers' comments:

Reviewer #1 (Remarks to the Author):

The data provided in the manuscript "A dual-lock inhibitory mechanism for fine-tuning enzyme activities of the HECT family E3 ligases" by Wang et al. give valuable insights into the regulation of HECT activity and are an important extension of previous studies, in which interactions between WW2 and HECT were analyzed on a structural level and in which the WW4-HECT interaction was analyzed biochemically. But so far no structural data on the latter interaction have been available. This study fills this gap and also extends the understanding of cancer-related mutations.

- Page 8: The authors state that "WW2L34 interacted with HECT more robustly than WW2L or LWW34 did". While this statement seems plausible, it does not appear to be supported by the referenced data. Figure 1B is an activity assay and Suppl. Fig. 1 B tests different constructs. It would thus be important to include WW2L34 in the binding assays in Fig. 1C.

The authors should also include a quantitative method, such as ITC, to determine the differences in binding between WW2L34 and WW2L and LWW34. Alternatively, if construct stability does not allow for these experiments, the authors should choose a weaker/non-quantitative wording when describing the differences in binding.

- Figure 2D: For the E2/E3 trans-thiolation assay a time course should be performed. Endpoint assays often disguise differences in this assay.

- Figure S1D: The authors observe considerable conformational changes. Beyond the mentioned 35° rotation (page 8, line 5) there is a 25°-rotation at the N-terminal part of the connector (L34HECT vs. 2L34HECT). This implies that different interfaces with the HECT domain are used. What are the surface areas of WW4, the connector, and WW2 with the HECT domain in either structure? Are these conformational changes provoked by crystal contacts? Is the 25° rotation the result of the missing WW2 domain? Do the authors attribute any functional significance to these states?

- For all structures, a discussion of crystal contacts and their possible influence/lack of influence on the discussed interfaces is missing.

- Fig. 4E: The levels of WT WWP1 clearly differ from those of H517Y, W549A and P651A. The authors should provide an explanation for that. For this purpose a cycloheximide chase assay and or addition of bortezomib should help to judge whether these hyper-active mutants promote their own degradation via auto-ubiquitination.

- The third structure for which statistics are provided in Table 1 does not appear anywhere in the text, except for Figure S1C. This structure should be compared in more detail to previously solved ones or - if no important conclusions can be drawn - the authors should consider moving into the supplements entirely or leaving it out.

Minor comments:

- Along with Figure 1A (or in the Methods section) it should be stated what sequence and length the linker has that connects L and the HECT domain in the constructs 2LHECT and LHECT. Likewise, information on how many residues bridge the gap between WW2 and WW3 in 234HECT is required.

- The comment in the caption of Figure S1C is confusing: "Note that in the reported WWP2 structure (PDB ID: 5TJ7), WW2L was directly fused with HECT, whereas in the structure in this study, WW2L was fused with HECT with a protease TEV-cutting site, which was then cleaved by protease TEV digestion." This suggests that the spacer bridging WW3 and 4 is even longer than in the case of the WWP2 structure reported before (PDB ID: 5TJ7). There is no other mention of the TEV site in the manuscript. What is the purpose of this site? Why was the construct cleaved?

- The choice of references for the roles of E3s as therapeutic targets (page 3) appears a bit random, as only Mund et al. is singled out as an original study on HECT inhibitors. It would be better to acknowledge the body of recent data on HECT inhibitors by citing e.g. Fajner et al (PMID 28771691) or Chen et al (PMID 30088849).
- The spelling of protein names is inconsistent throughout the text, e.g. Nedd4 versus NEDD4L
- The domains in Fig. 2C should be labeled according to Figs 2A & 2B.
- Fig. S1D illustrates the binding of the HECT N-terminus to WW4 and its similarity to a typical WW-NPXY interaction. An additional panel in which this interaction is shown for both L34 and 2L34 would be helpful to strengthen the comment that WW4 binds to the same L site on the HECT domain in the two structures (page 8, line 5).
- Fig. 3A: Which interface is shown here, that of 2L34HECT or that of L34HECT?
- Page 14: The names of mutated constructs, such as Y491AHECT are misleading, since they appear to imply that these mutations were in the context of the HECT domain. However, in Figs 5E and 5J these variants were tested in the context of 12L34HECT (judged based on protein size). Please indicate unambiguously in which construct context the mutations were tested.
- Figure S1B. The nomenclature here is very confusing (e.g. "WW" is now the longest construct...). Please match the construct names with the names used in the remainder of the text.
- Table 1: The authors chose rather conservative criteria for cutting back the data. I/sigI are rather high. Please quote CC1/2 values.

Reviewer #2 (Remarks to the Author):

The manuscript is a follow-up of three recent publications (Chen et al., *Mol. Cell*, 2017; Zhu et al., *EMBO rep.*, 2017; Yao et al., *JBC*, 2018) that explore the auto-inhibition of the highly homologous Nedd4-family ubiquitin ligases WWP1, WWP2, Itch and Su(dx). Nedd4-family ligases regulate various signal transduction pathways and share a common C2-WW-HECT domain architecture that enables them to adopt an auto-inhibited state in the absence of signaling cues. Whether and which mechanisms of C2-WW-HECT inhibition are shared within HECT-type ligases is poorly understood. In this manuscript, the authors solved the crystal structures of two WWP1 fragments (WW2LWW34HECT and WW2LHECT) and a WWP2 fragment (WW2LHECT) that has already been published (Chen et al., *Mol. Cell*, 2017). While the presented structures are highly similar to each other and to known structures, the authors uncover novel interactions between the WW4 and the HECT domains in WWP1. The authors show in functional assays that full inhibition of WWP1, WWP2 and Itch requires both the WW2L region and the WW4 domain. Moreover, cancer related mutations at the WW2-, L- and WW4-interface enhance WWP1 activity and promote cell migration and substrate ubiquitination. Lastly, the authors provide evidence that in Nedd4 (as already shown in Chen et al., *Mol. Cell*, 2017) and Smurf ligases the WW1-WW2-linker may contribute to auto-inhibition in a manner similar to WWP1, WWP2 and Itch.

Overall, the discovery of an additional WW binding surface on the HECT domain and its involvement in auto-inhibition adds an important piece of information as to how HECT-type ligases are regulated and which mechanistic aspects are conserved within this important class of ubiquitin ligases. However, although the structural and functional analyses are overall convincing, the study suffers from redundancies with previous studies (Chen et al., *Mol. Cell*, 2017; Zhu et al., *EMBO rep.*, 2017; Yao et al., *JBC*, 2018) and should rather focus on the findings that are novel and substantiate these in order for this study to be suitable for publication in *Nature Communications*.

Major issues:

- The structural differences between the 2L34HECT and L34HECT constructs should be explored in more detail. Is the rotation of the WW4 functionally relevant or a crystallization artifact? Are crucial WW4-HECT contacts in the 2L34HECT structure preserved in the L34HECT structure? The assays in Fig. 3B, 4B and Suppl. Fig. 2 should be performed with the L34HECT fragment to analyze the relevance of the WW4:HECT interface in the L34HECT structure.
- The term "semi-open state" for the L34HECT structure is problematic and would suggest an increase in conformational dynamics or a major conformational change of the HECT lobes that would explain the enhanced activity of the L34HECT fragment. However, this is not observed in the crystal structures. The term "activated" may be more appropriate.
- The term "dual-lock" does not seem appropriate since HECT inhibition requires at least three structural elements: the linker, the WW2 and the WW4 domain. Also, although a headset may be fair analogy the abbreviations "L" and "C" are confusing as they also refer to the linker and the C-lobe.
- The E697K cancer mutation is located remote from the lobe interface or the 2L34 interface. How do the authors explain the increased activity of this mutant?
- The authors should show uncropped gel images in the Supplement and clearly mark gels that ran separately (e.g. does Fig. 5G show a single gel?).
- A role of the WW1-WW2-linker in Nedd4 auto-inhibition has already been reported (Chen et al., Mol. Cell, 2017). Moreover, if the Nedd4/Smurf2 WW1 domain would bind to the same site as the WW4 domain in WWP1, then the orientation of the linker helix in Nedd4 would be reversed as compared to WWP1/2 and Itch, since at least in Nedd4 the linker is directly C-terminal of the WW1 domain. This should be taken into account and discussed in the text.
- Fig. 6C: The position of the helical linker region (L) in Smurf2 is incorrect. In contrast to Nedd4, there are 20 aa separating the WW1 domain from the helical linker region in Smurf2. These differences should also be discussed in the text.
- Smurf1 lacks the WW1 domain (its WW domains correspond to WW2 and WW3 in Smurf2), and the sequence shown in Fig. 6E is only poorly conserved in Smurf1. This does not fit to an inhibition mechanism that is conserved among Smurf1, Smurf2, Nedd4 and Nedd4L. The authors should thus revise the interpretations of their results for Smurf1 or perform additional experiments to show which WW domain in Smurf1 may occupy the WW4 binding surface in WWP1.

Minor issues:

- The text needs a thorough editing of the English language.
- Abbreviations should be defined (e.g. 2L34HECT etc. on p. 5)
- Fig. 3E, 4B, 6B and 6D: The authors should refer to the corresponding supplementary figures.
- Fig. 4: Which structure is shown in (A)? What do the asterisks refer to?
- Fig. 6E: The sequences shown in the alignment do not match with the numbers given on the sides.
- Materials and methods should be described in more detail.
- The authors should provide a table of all protein constructs used in this study
- References should be formatted and placed properly.

Reviewer #3 (Remarks to the Author):

In this manuscript, Wang et. al report crystal structures of the HECT E3 ubiquitin ligase WWP1 in fully inhibited 'closed' (2L34HECT) and partially inhibited 'semi-open' conformations (L34HECT). Similar to previous studies on WWP2 and Itch E3 ligases, contacts between WW2/L and the HECT domain that contribute to autoinhibition are observed the closed WWP1 structure. In both the closed and semi-open WWP1 structures, a previously unobserved interface between the WW4 and HECT domains is observed that also contributes to autoinhibition. The authors performed extensive in vitro experiments that support a model in which the bilateral interaction between the WW2 and WW4 domains with the R and L sites of the HECT domain, along with the interaction of the WW2-WW3 linker and C site of the HECT domain serve as a dual-lock autoinhibitory mechanism of E3 ligase activity. The authors extend their structural and biochemical studies into cells and show that mutations in observed in cancers occurring at residues within the WW2L34 region of WWP1 increase ligase activity. The authors also show that these mutations promote cell migration and proliferation likely through downregulation of p63. Finally, the authors show that the new WW4/HECT interaction at the L site is likely conserved in WWP2 and Itch and that a variant dual-lock mechanism is likely present in Nedd4 and Smurf.

Overall, this is a carefully performed comprehensive study that provides significant new insights into the regulation of a major family of HECT E3 ubiquitin ligases. In my opinion this manuscript would be suitable for publication in Nature Communications if the below issues are adequately addressed.

Major comments:

* The authors should address why the WW4/HECT interaction was not observed in the Itch structure (PDB:5XMC) despite the fact that WW3/4 are present in the construct?

* Is the semi-open structure relevant? In the L34HECT structure it is reported that the WW4 is rotated by 35 degrees compared to the 2L34HECT structure. Presumably, this alters the network of contacts between WW4 and the HECT domain that are described in Figure 3A based on the 2L34HECT structure and which are probed through structure-function studies throughout the rest of the manuscript. The difference in the WW4-HECT interactions between the semi-open and closed structures should be addressed. Have the authors performed any experiments to test whether interactions between the WW4 and HECT domains unique to the semi-open structure are mechanistically important? I think this is an important issue that needs to be addressed. Is the WW4 domain involved in crystal contacts in the semi-open and/or closed structures?

Minor comments:

* In my opinion the abstract is too detailed and the authors should consider making an effort to summarize their findings in a way that will appeal to a broad audience

* There are many typographical errors throughout the manuscript, particularly the discussion. The authors should very carefully review their text to address these typos. Some examples:

Line 17 of page 15 - 'trails'

Please clarify lines 18-20 of page 15

Perhaps consider a word other than 'destroyed' for line 16 page 16

Line 11 page 18 has an endnote error

Lines 13-16 on page 18 are very confusing and should be clarified

Line 19 page 18- typo 'Ndifip1'

Line 21 page 18- typo? 'after T activation'

* Sentence 1 of the introduction is not accurate (E2s determine linkage-type for RING E3s)

* On page 6, the authors invoke the idea of closed and semi-open states to explain their biochemical data. This precedes description of the structures. Particularly problematic is lines 15-17, 'Both 2LHECT and L34HECT mutants possess a slightly elevated ligase activity compared to L234HECT, indicating that they adopt an incompletely inhibited semi-open state'.

* Line 21 page 6- define 'Trx'

* Line 9 page 18- referring to Y543E as a phosphomimetic is probably a stretch

(Our responses to the reviewers' comments are shown in italics and highlighted in blue):

Reviewer #1:

1) Page 8: The authors state that “WW2L34 interacted with HECT more robustly than WW2L or LWW34 did”. While this statement seems plausible, it does not appear to be supported by the referenced data. Figure 1B is an activity assay and Suppl. Fig. 1 B tests different constructs. It would thus be important to include WW2L34 in the binding assays in Fig. 1C. The authors should also include a quantitative method, such as ITC, to determine the differences in binding between WW2L34 and WW2L and LWW34. Alternatively, if construct stability does not allow for these experiments, the authors should choose a weaker/non-quantitative wording when describing the differences in binding.

We thank the reviewer for noting this issue for us, and have included WW2L34 in the pull-down assay in Supplementary Fig. 1b (presented below for convenience of viewing). Consistent with our model, WW2L34 could pull down more HECT than WW2L or LWW34 did. As the reviewer pointed out, the protein behavior of WW2L is not suitable for a quantitative measurement, so we modified the description of the binding differences in the text as the reviewer suggested.

Supplementary Fig.1B, GST pull-down assay of GST-tagged WWP1 WW, WW1, WW2L34, WW2L or LWW34 with Trx-HECT.

2) Figure 2D: For the E2/E3 transthiolation assay a time course should be performed. Endpoint assays often disguise differences in this assay.

Following the reviewer's suggestion, we have repeated the E2/E3 transthiolation assay in Fig. 2d (presented below for convenience of viewing). The data suggests that though L is essential for keeping WWP1 in its fully inhibited state, the isolated L is not sufficient for locking HECT in the inactive state.

Fig. 2d, E2-E3 transthiolation assay of WWP1 C terminal five residue-removed constructs 2L34HECTΔCT, 234HECTΔCT, LHECTΔCT and HECTΔCT.

3) Figure S1D: The authors observe considerable conformational changes. Beyond the mentioned 35° rotation (page 8, line 5) there is a 25°-rotation at the N-terminal part of the connector (L34HECT vs. 2L34HECT). This implies that different interfaces with the HECT domain are used. What are the surface areas of WW4, the connector, and WW2 with the HECT domain in either structure? Are these conformational changes provoked by crystal contacts? Is the 25° rotation the result of the missing WW2 domain? Do the authors attribute any functional significance to these states?

We thank the reviewer for noting this issue for us, and have carefully analyzed both structures. The interaction interfaces are identical between WW4 and HECT in L34HECT and 2L34HECT (Figure 3a and Supplementary Fig. 1f, presented below for convenience of viewing). Our assumption is that the WW4-binding N-terminal extension of HECT is a flexible loop, which allows HECT-bound WW4 to rotate a certain extent upon crystal packing with a symmetric molecule in 2L34HECT (Supplementary Fig. 1g, presented below for convenience of viewing). We think the 25° rotation of the N-terminal part of L which has few packing with HECT, is the result of the missing WW2 domain and crystal contacts (Fig. 1 and Supplementary Fig. 1g). The interaction surfaces are identical between the central part of L and HECT in both structures, which provide the driven force for L-HECT interaction. As the crucial WW4-HECT and L-HECT contacts in the 2L34HECT structure preserve in the L34HECT structure, we believe these crystal contacts induced conformational changes have no functional significance.

Due to the lack of WW2-HECT interaction, LWW34-mediated inhibition on HECT in L34HECT should be weaker than WW2L34 does in 2L34HECT (Fig. 1). As a result, when E2-Ub attacks the active Cys of HECT, it will be easier for HECT in L34HECT to undergo the transition from the inactive T-shape to a catalytically active L-shape conformation. Note that the isolated HECT domains in Itch and WWP1/2 all adopt a closed T-shape conformation based on their crystal structures (PDB IDs: 3TUG, 1ND7, 4Y07). Nevertheless, all these isolated HECT domains show a strong ligase activity when incubated with E2-Ub. We assume that HECT domains from Itch and WWP1/2, either in the fully active (HECT), partially active (L34HECT), or fully inactive (2L34HECT) state are prone to adopt the inactive T-shape conformation during crystallization.

Fig. 3a, The detailed structure of WW2-HECT, L-HECT and WW4-HECT interfaces from 2L34HECT.

Supplementary Fig. 1f, The detailed structure of L-HECT and WW4-HECT interfaces from L34HECT.

Supplementary Fig. 1g, Crystal contacts of 2L34HECT in the crystal. Left panel: M414 from L (magenta) of 2L34HECT interacts with N745 of a symmetric molecule (grey). Right panel: The main chains of T519 and R520 from WW4 (orange) of 2L34HECT form hydrogen bonds with R601 from a symmetric molecule (grey). E509 from WW4 (orange) forms a salt bridge with K531 from a symmetric molecule (grey). All these crystal contacts do not exist in the L34HECT (blue) structure.

Fig. 1, Interactions between the N-terminal part of L and HECT in 2L34HECT and L34HECT.

4) For all structures, a discussion of crystal contacts and their possible influence/lack of influence on the discussed interfaces is missing.

Following this reviewer's suggestion, we have added the description and discussion of crystal contacts in the revised manuscript (page 9, line 20).

5) Fig. 4E: The levels of WT WWP1 clearly differ from those of H517Y, W549A and P651A. The authors should provide an explanation for that. For this purpose a cycloheximide chase assay and or addition of bortezomib should help to judge whether these hyper-active mutants promote their own degradation via auto-ubiquitination.

Following the reviewer's suggestion, we repeated the experiment in Fig. 4e in the presence of MG-132, another proteasome inhibitor like bortezomib (presented below for convenience of viewing). When treated with MG-132, the protein levels of WWP1 WT and mutants (H517Y, W549A and P651A) were comparable, implying that these hyper-active mutants indeed promoted their own degradation via auto-ubiquitination. In line with this result, when treated

with MG-132, the levels of endogenous Δ Np63 α dramatically increased in MCF-10A cells expressing WWP1 mutants, further demonstrating that these WWP1 mutants promote degradation of Δ Np63 α through ubiquitin - proteasome pathway.

Fig. 4e, Western blotting analyses of Δ Np63 α level in MCF-10A cells stably expressing WWP1 WT or mutants with or without treatment of MG-132.

6) The third structure for which statistics are provided in Table 1 does not appear anywhere in the text, except for Figure S1C. This structure should be compared in more detail to previously solved ones or - if no important conclusions can be drawn - the authors should consider moving into the supplements entirely or leaving it out.

Indeed there is no significant difference between the WWP2 2LHECT structure and the previous one. We have removed the data in the revised manuscript.

7) Along with Figure 1A (or in the Methods section) it should be stated what sequence and length the linker has that connects L and the HECT domain in the constructs 2LHECT and LHECT. Likewise, information on how many residues bridge the gap between WW2 and WW3 in 234HECT is required.

We thank the reviewer for the suggestion. We have added the information of all the constructs in Supplementary Table 1.

8) The comment in the caption of Figure S1C is confusing: "Note that in the reported WWP2 structure (PDB ID: 5TJ7), WW2L was directly fused with HECT, whereas in the structure in this study, WW2L was fused with HECT with a protease TEV-cutting site, which was then cleaved by protease TEV digestion." This suggests that the spacer bridging WW3 and 4 is even longer than in the case of the WWP2 structure reported before (PDB ID: 5TJ7). There is no other mention of the TEV site in the manuscript. What is the purpose of this site? Why was the construct cleaved?

To avoid artificial packing by directly fusing WWP2 WW2L with HECT (in real situation these two parts are separated by WW34, ~70 aa), we added a protease TEV-cutting site between WW2L and HECT, and then digested this site with protease TEV. The resulted protein were then used for crystallization. Indeed, there is no significant difference between this WWP2 2LHECT structure and the previous one. We have removed the data in the revised manuscript.

9) The choice of references for the roles of E3s as therapeutic targets (page 3) appears a bit random, as only Mund et al. is singled out as an original study on HECT inhibitors. It would be better to acknowledge the body of recent data on HECT inhibitors by citing e.g. Fajner et al (PMID 28771691) or Chen et al (PMID 30088849).

Following the reviewer's suggestion, we have cited these papers in the revised manuscript.

10) The spelling of protein names is inconsistent throughout the text, e.g. Nedd4 versus NEDD4L

We have modified the text following this reviewer's comment.

11) The domains in Fig. 2C should be labeled according to Figs 2A & 2B.

We have modified Fig. 2c following this reviewer's comment.

12) Fig. S1D illustrates the binding of the HECT N-terminus to WW4 and its similarity to a typical WW-NPXY interaction. An additional panel in which this interaction is shown for both L34 and 2L34 would be helpful to strengthen the comment that WW4 binds to the same L site on the HECT domain in the two structures (page 8, line 5).

Following this reviewer's suggestion, we have provided the information in Supplementary Fig. 1f (see point 3 above for more details).

13) Fig. 3A: Which interface is shown here, that of 2L34HECT or that of L34HECT?

We have clarified the description in the revised manuscript (see point 3 above for more details).

14) Page 14: The names of mutated constructs, such as Y491AHECT are misleading, since they appear to imply that these mutations were in the context of the HECT domain. However, in Figs 5E and 5J these variants were tested in the context of 12L34HECT (judged based on protein size). Please indicate unambiguously in which construct context the mutations were tested.

Sorry for that, and we have clarified the description in the revised manuscript.

15) Figure S1B. The nomenclature here is very confusing (e.g. "WW" is now the longest construct...). Please match the construct names with the names used in the remainder of the text.

Sorry for the mistake, and we have modified the construct names.

16) Table 1: The authors chose rather conservative criteria for cutting back the data. I/sigI are rather high. Please quote CC1/2 values.

Following the reviewer's suggestion, we have quoted CC1/2 values in Table 1.

Reviewer #2:

1) The structural differences between the 2L34HECT and L34HECT constructs should be explored in more detail. Is the rotation of the WW4 functionally relevant or a crystallization artifact? Are crucial WW4-HECT contacts in the 2L34HECT structure preserved in the L34HECT structure? The assays in Fig. 3B, 4B and Suppl. Fig. 2 should be performed with the L34HECT fragment to analyze the relevance of the WW4:HECT interface in the L34HECT structure.

We thank the reviewer for noting this issue for us, and have carefully analyzed both structures. The interaction interfaces are identical between WW4 and HECT in L34HECT and 2L34HECT (Figure 3a and Supplementary Fig. 1f, presented below for convenience of viewing). Our assumption is that the WW4-binding N-terminal extension of HECT is a flexible loop, which allows HECT-bound WW4 to rotate a certain extent upon crystal packing with a symmetric molecule in 2L34HECT (Supplementary Fig. 1g, presented below for convenience of viewing). As the crucial WW4-HECT contacts in the 2L34HECT structure preserve in the L34HECT structure, we believe the rotation of WW4 has no functional significance.

Due to the lack of WW2-HECT interaction, LWW34-mediated inhibition on HECT in L34HECT should be weaker than WW2L34 does in 2L34HECT (Fig. 1). As a result, when E2-Ub attacks the active Cys of HECT, it will be easier for HECT in L34HECT to undergo the transition from the inactive T-shape to a catalytically active L-shape conformation. Note that the isolated HECT domains in Itch and WWP1/2 all adopt a closed T-shape conformation based on their crystal structures (PDB IDs: 3TUG, 1ND7, 4Y07). Nevertheless, all these isolated HECT domains show a strong ligase activity when incubated with E2-Ub. We assume that HECT domains from Itch and WWP1/2, either in the fully active (HECT), partially active (L34HECT), or fully inactive (2L34HECT) state are prone to adopt the inactive T-shape conformation during crystallization.

We agree with the reviewer that biochemical and functional validation of the WW4:HECT interface will be more straightforward using the L34HECT fragment. However, to compare the contribution of both WW2-HECT and WW4-HECT interactions on WWP1 auto-inhibition, we chose to perform the experiments with the fragments containing both WW2 and WW4.

Fig. 3a, The detailed structure of WW2-HECT, L-HECT and WW4-HECT interfaces from 2L34HECT.

Supplementary Fig. 1f, The detailed structure of L-HECT and WW4-HECT interfaces from L34HECT.

Supplementary Fig. 1g, Crystal contacts of 2L34HECT in the crystal. Left panel: M414 from L (magenta) of 2L34HECT interacts with N745 of a symmetric molecule (grey). Right panel: The main chains of T519 and R520 from WW4 (orange) of 2L34HECT form hydrogen bonds with R601 from a symmetric molecule (grey). E509 from WW4 (orange) forms a salt bridge with K531 from a symmetric molecule (grey). All these crystal contacts do not exist in the L34HECT (blue) structure.

2) The term “semi-open state” for the L34HECT structure is problematic and would suggest an increase in conformational dynamics or a major conformational change of the HECT lobes that would explain the enhanced activity of the L34HECT fragment. However, this is not observed in the crystal structures. The term “activated” may be more appropriate.

We thank the reviewer for the suggestion. We have modified the description in revised manuscript.

3) The term “dual-lock” does not seem appropriate since HECT inhibition requires at least three structural elements: the linker, the WW2 and the WW4 domain. Also, although a headset may be fair analogy the abbreviations “L” and “C” are confusing as they also refer to the linker and the C-lobe.

In the revised manuscript, we have changed the term “dual-lock” to be “multi-lock”, and denoted “Right ear” as “Re”, “Left ear” as “Le”, and the connecting Headband as “H”.

4) The E697K cancer mutation is located remote from the lobe interface or the 2L34 interface. How do the authors explain the increased activity of this mutant?

In our previous work, we found that WW2L binding induced allosteric conformational change of Itch HECT, leading to formation of the H830-E639 hydrogen bond, which was important for keeping Itch in its closed conformation (Zhu, EMBO Report, 2017). Interestingly, the cancer-related substitution of the evolutionarily conserved Glu697 (corresponding to E639 in Itch) to Lys resulted in significant increase of WWP1 auto-ubiquitination (Fig. 4b), implying that Glu697 may be involved in the auto-inhibition of WWP1, possibly through forming the

same His/Glu hydrogen bond (H830 in Itch is also conserved in WWP1).

5) The authors should show uncropped gel images in the Supplement and clearly mark gels that ran separately (e.g. does Fig. 5G show a single gel?).

The Fig. 5g includes two gels. We have modified Fig. 5g to clearly mark gels that ran separately and provided uncropped gel images in the Source Data.

6) A role of the WW1-WW2-linker in Nedd4 auto-inhibition has already been reported (Chen et al., Mol. Cell, 2017). Moreover, if the Nedd4/Smurf2 WW1 domain would bind to the same site as the WW4 domain in WWP1, then the orientation of the linker helix in Nedd4 would be reversed as compared to WWP1/2 and Itch, since at least in Nedd4 the linker is directly C-terminal of the WW1 domain. This should be taken into account and discussed in the text.

Indeed, according to our structural model, the orientation of the linker helix in Nedd4 is reversed as compared to WWP1/2 and Itch. Following the reviewer's suggestion, we have discussed this issue in the revised manuscript (page 16, line 10).

7) Fig. 6C: The position of the helical linker region (L) in Smurf2 is incorrect. In contrast to Nedd4, there are 20 aa separating the WW1 domain from the helical linker region in Smurf2. These differences should also be discussed in the text.

The labelling of the helical linker region was according to the structural prediction of Nedd4 and Nedd4L though PSIPRED 3.3, and we could not predict a helix in the same region of Smurf2. Based on our structural modelling of Nedd4 and biochemical analyses of Nedd4/4L and Smurf2, we proposed the most possible interaction mode between L and HECT in these E3s in Fig. 6e and 6g. We have mentioned the ~30 aa insertion between WW1 and the helical linker region in Smurf2 in the revised manuscript (page 16, line 5).

8) Smurf1 lacks the WW1 domain (its WW domains correspond to WW2 and WW3 in Smurf2), and the sequence shown in Fig. 6E is only poorly conserved in Smurf1. This does not fit to an inhibition mechanism that is conserved among Smurf1, Smurf2, Nedd4 and Nedd4L. The authors should thus revise the interpretations of their results for Smurf1 or perform additional experiments to show which WW domain in Smurf1 may occupy the WW4 binding surface in WWP1.

Our data in Supplementary Fig. 6a showed that the activity of Smurf1 increased upon deletion of WW domains, implying that WW domains may be involved in auto-inhibition. However, as the reviewer pointed out, the sequence shown in Fig. 6e is not conserved in Smurf1. Unfortunately, due to the poor protein behavior, we could not further characterize the detailed molecular mechanism of Smurf1 autoinhibition. We have modified our interpretations on Smurf1 data.

9) The text needs a thorough editing of the English language.

We have modified the manuscript.

10) Abbreviations should be defined (e.g. 2L34HECT etc. on p. 5)

We have defined the Abbreviations of our constructs in Figs. 1a, 5a, 5f, 6a, 6c, Supplementary Fig. 6c and Supplementary Table1.

11) Fig. 3E, 4B, 6B and 6D: The authors should refer to the corresponding supplementary figures.

We have modified the Figure legends.

12) Fig. 4: Which structure is shown in (A)? What do the asterisks refer to?

The structure is WWPI 2L34HECT. The asterisks represent cancer-related mutants. The relative information has been provided in Figure legends.

13) Fig. 6E: The sequences shown in the alignment do not match with the numbers given on the sides.

In this study, we used human Nedd4 isoform 3 (uniprot identifier P46934-3, aa 1-900), human Nedd4L Isoform 1 (identifier Q96PU5-1, aa 1-975) and human Smurf2 (uniprot identifier Q9HAU4-1, aa 1-748) to align the linker region (L). The numbers have been labelled in Fig. 6e.

14) Materials and methods should be described in more detail.

Following the reviewer's suggestion, we have modified the Materials and methods.

15) The authors should provide a table of all protein constructs used in this study

We have provided the information in the Supplementary Table 1.

16) References should be formatted and placed properly.

We have reformatted the references.

Reviewer #3:

1) The authors should address why the WW4/HECT interaction was not observed in the Itch structure (PDB:5XMC) despite the fact that WW3/4 are present in the construct?

Though the WW4-binding “IAY” motif from the N-terminal extension of WWP1 HECT is completely conserved in Itch, there are still some variations in the WW4-HECT packing surface. For example, Trp549^{HECT} which interacts with WW4 in both WWP1 2L34HECT and L34HECT (Fig. 3a and Supplementary 1f), is substituted with Ala in Itch, that may weaken the WW4-HECT interaction. Moreover, when we superimpose WWP1 2L34HECT structure to Itch structure (PDB:5XMC), though WW2-L-HECT in both structures fit very well (Fig. 1c), the “Le” site in Itch is occupied by the WW2 domain of a symmetric molecule (Fig. II). Such competition between a symmetric Itch’ WW2 and WW4 towards HECT during crystallization may be a reason why the WW4/HECT interaction was not observed in the crystal structure of Itch 12L34HECT.

Fig. II, Crystal contacts of Itch 2L34HECT (blue). The “Le” site in Itch is occupied by the WW2 domain of a symmetric molecule (grey).

2) Is the semi-open structure relevant? In the L34HECT structure it is reported that the WW4 is rotated by 35 degrees compared to the 2L34HECT structure. Presumably, this alters the network of contacts between WW4 and the HECT domain that are described in Figure 3A based on the 2L34HECT structure and which are probed through structure-function studies throughout the rest of the manuscript. The difference in the WW4-HECT interactions between the semi-open and closed structures should be addressed. Have the authors performed any experiments to test whether interactions between the WW4 and HECT domains unique to the semi-open structure are mechanistically important? I think this is an important issue that needs to be addressed. Is the WW4 domain involved in crystal contacts in the semi-open and/or closed structures?

We thank the reviewer for noting this issue for us, and have carefully analyzed both structures. The interaction interfaces are identical between WW4 and HECT in L34HECT and 2L34HECT (Figure 3a and Supplementary Fig. 1f, presented below for convenience of viewing). Our assumption is that the WW4-binding N-terminal extension of HECT is a flexible loop, which allows HECT-bound WW4 to rotate a certain extent upon crystal packing with a symmetric molecule in 2L34HECT (Supplementary Fig. 1g, presented below for convenience of viewing). As the crucial WW4-HECT contacts in the 2L34HECT structure preserve in the L34HECT structure, we believe the rotation of WW4 has no functional significance.

Due to the lack of WW2-HECT interaction, LWW34-mediated inhibition on HECT in

L34HECT should be weaker than *WW2L34* does in *2L34HECT* (Fig. 1). As a result, when *E2-Ub* attacks the active Cys of *HECT*, it will be easier for *HECT* in *L34HECT* to undergo the transition from the inactive *T*-shape to a catalytically active *L*-shape conformation. Note that the isolated *HECT* domains in *Itch* and *WWP1/2* all adopt a closed *T*-shape conformation based on their crystal structures (PDB IDs: 3TUG, 1ND7, 4Y07). Nevertheless, all these isolated *HECT* domains show a strong ligase activity when incubated with *E2-Ub*. We assume that *HECT* domains from *Itch* and *WWP1/2*, either in the fully active (*HECT*), partially active (*L34HECT*), or fully inactive (*2L34HECT*) state are prone to adopt the inactive *T*-shape conformation during crystallization.

Fig. 3a, The detailed structure of *WW2-HECT*, *L-HECT* and *WW4-HECT* interfaces from *2L34HECT*.

Supplementary Fig. 1f, The detailed structure of *L-HECT* and *WW4-HECT* interfaces from *L34HECT*.

Supplementary Fig. 1g, Crystal contacts of *2L34HECT* in the crystal. Left panel: *M414* from *L* (magenta) of *2L34HECT* interacts with *N745* of a symmetric molecule (grey). Right panel: The main chains of *T519* and *R520* from *WW4* (orange) of *2L34HECT* form hydrogen bonds with *R601* from a symmetric molecule (grey). *E509* from *WW4* (orange) forms a salt bridge with *K531* from a symmetric molecule (grey). All these crystal contacts do not exist in the *L34HECT* (blue) structure.

3) In my opinion the abstract is too detailed and the authors should consider making an effort to summarize their findings in a way that will appeal to a broad audience

Following the reviewer's suggestion, we have modified the abstract.

4) There are many typographical errors throughout the manuscript, particularly the discussion. The authors should very carefully review their text to address these typos. Some examples:

Line 17 of page 15 - 'trails'

Please clarify lines 18-20 of page 15

Perhaps consider a word other than 'destroyed' for line 16 page 16

Line 11 page 18 has an endnote error

Lines 13-16 on page 18 are very confusing and should be clarified

Line 19 page 18- typo 'Ndifip1'

Line 21 page 18- typo? 'after T activation'

Sorry for that, and we have corrected the typographical errors.

5) Sentence 1 of the introduction is not accurate (E2s determine linkage-type for RING E3s)

We have modified this sentence.

6) On page 6, the authors invoke the idea of closed and semi-open states to explain their biochemical data. This precedes description of the structures. Particularly problematic is lines 15-17, 'Both 2LHECT and L34HECT mutants possess a slightly elevated ligase activity compared to L234HECT, indicating that they adopt an incompletely inhibited semi-open state'.

We have modified the manuscript.

7) Line 21 page 6- define 'Trx'

We have defined 'Trx' in the revised manuscript.

8) Line 9 page 18- referring to Y543E as a phosphomimetic is probably a stretch

Following the reviewer's suggestion, we have modified the manuscript.

Reviewers' comments:

Reviewer #1 (Remarks to the Author):

The authors have addressed all the points carefully, so I consider the manuscript ready for publication.

Reviewer #2 (Remarks to the Author):

The major novelty of the manuscript is the visualization of the WW4 domain as an additional inhibitory element in the Nedd4-family members WWP1, WWP2 and Itch. Yet, although all reviewers have demanded more detailed insights into the functional relevance of the WW4-HECT interaction, the authors provide only speculations instead of actual additional experiments that would address this issue. As such the manuscript requires further revision and a more detailed analysis of the WW4-HECT interaction to be suitable for publication.

Other points that have not been properly addressed:

- Fig. 1b, 4f, 5b, 5g, Suppl. Fig. 6b,d,e,f, g: The authors should not merge gels that were run separately, i.e. gels should be boxed individually and molecular weight markers should be provided for the separate gels. E.g. the molecular weight marker shown in Fig. 1b belongs to the left gel, but not to the center or right hand gel. Molecular weights are not indicated in Fig. 4f.
- Source Data: Fig. 4e,g are not shown as uncropped gels. Molecular weight markers are missing.
- Fig. 6e: The numbering of the sequences on the left and right of the alignment in Fig. 6e still does not match the sequences shown. E.g. for Nedd4 a sequence stretch from residue 223-347 would refer to 124 aa being displayed in the alignment. What do the numbers in angle brackets refer to? The authors should display the residue numbers of the sequences that are actually shown in the alignment.
- Fig. 6c/e: The Smurf2 sequence shown in Fig. 6e corresponds to residues 217-249 (Uniprot identifier Q9HAU4-1, aa 1-748) (and does not start at residue 189, see comment above). This means that, in contrast to Nedd4 and Nedd4L, the linker region in Smurf2 starts 26 aa C-terminal of the WW1 domain and lies directly N-terminal of the WW2 domain. The authors should modify Fig. 6c accordingly.
- p23-24, protein purification: The authors should either refer to protease 3C or Precision protease or make clear that the two names refer to the same protein. It would be nice to point out that GST-tagged proteins were only used for pull-down assays and all other studies (except substrate ubiquitination) were performed with proteins expressed from pET-32a vectors.
- p24, lines 5-8 and p25, line 17 are contradictory.
- p25, line 18, 22: The concentration of the E3s and DTT is missing.
- p26, line 16: The authors should state the buffer conditions in which the proteins were concentrated. Which concentration refers to which protein? Please state the exact concentrations that were used for crystallization.
- p26, line 19-21: Which reservoir solution and cryo-condition belongs to which construct?
- p29, line 11,12: This is not a proper sentence.

Reviewer #3 (Remarks to the Author):

In their revised manuscript, Wang et. al have made significant improvements in response to the reviewer comments. In their rebuttal letter, the authors have suitably addressed reviewer concerns regarding crystal packing in their structures and a failure to observe the 'multi-lock' autoinhibited form of Itch in a previously determined structure (PDB:5XMC). However, much this information has not been incorporated into the revised manuscript and its inclusion in an additional revision would add to the clarity and scholarliness of the manuscript. I fully support publication of this manuscript in Nature Communications should the following minor issues be addressed:

* In the section titled 'The multi-lock autoinhibition mode exists in WWP2 and Itch', the authors should acknowledge that the multi-lock autoinhibited state of Itch (with WW4/HECT interactions) was not observed in the crystal structure of Itch in PDB:5XMC despite the fact that WW3/4 were present in the construct and provide some of the discussion they included in their rebuttal to rationalize this apparent contradiction.

*In the section 'Overall structures of the inactive and partially active WWP1' the authors should include information on crystal packing that they included in their response to the same inquiry from all three reviewers. As it is written now, the described 35 degree rotation of WW4 and 25 degree rotation of L appear to be significant, whereas the authors have concluded in their rebuttal that they are not significant and are likely due to crystal packing. Likewise, in the section 'The WW2L34-HECT interface' on page 9 the authors should point out that while one might expect that the network of interactions at the WW4-HECT interface to be different due to the 35 degree rotation of WW4, that the network of contacts is actually similar. They should also include the explanation provided in their rebuttal for how the network of contacts remains similar despite the WW4 rotation.

*On page 18, the authors still refer to the Y543E mutant as a phosphomimetic. Apart from harboring a negative charge, glutamate looks nothing like phosphotyrosine as it is far shorter and lacks an aromatic ring. In the third from last sentence on this page, the authors should replace 'the phosphomimetic mutant' with a phrase along the lines of 'introduction of a negative charge at Tyr543 in the form of a Y543E mutation' and should acknowledge the fact that glutamate and phosphotyrosine are structurally different which introduces potential caveats in interpreting results obtained with the mutant as they pertain to insights on phosphorylation.

(Our responses to the reviewers' comments are shown in italics and highlighted in blue):

Reviewer #1 (Remarks to the Author):

The authors have addressed all the points carefully, so I consider the manuscript ready for publication.

We thank Reviewer #1 for his/her final approval.

Reviewer #2 (Remarks to the Author):

The major novelty of the manuscript is the visualization of the WW4 domain as an additional inhibitory element in the Nedd4-family members WWP1, WWP2 and Itch. Yet, although all reviewers have demanded more detailed insights into the functional relevance of the WW4-HECT interaction, the authors provide only speculations instead of actual additional experiments that would address this issue. As such the manuscript requires further revision and a more detailed analysis of the WW4-HECT interaction to be suitable for publication.

Following the reviewer's suggestion, we have conducted biochemical and functional validation of the WW4:HECT interface using the L34HECT fragment. The point mutation at the WW4-HECT binding surface ($H517A^{WW4}$ or $Y543A^{HECT}$) severely impaired the WW4-HECT interaction (Supplementary Fig. 2b, presented below for convenience of viewing). Meanwhile, mutations at the WW4-HECT packing interface in WW4 ($E503A$, $H517A$, and $H517Y$) or HECT ($Y543A$, $Y543E$, $W549A$) all led to significantly elevated autoubiquitination of WWP1 L34HECT (Supplementary Fig. 2c). Overall, the above results further highlight the critical role of WW4-HECT interaction in keeping WWP1 in the inactive state.

Supplementary Fig. 2b, GST pull-down assay of WWP1 GST-LWW34 with Trx-HECT. Mutations that were predicted to impair interactions at the "Le" site in WW4-HECT ($Y543A^{HECT}$ and $H517A^{WW4}$) disrupted the interaction. 2c, In vitro autoubiquitination assay of various WWP1 L34HECT mutants.

- Fig. 1b, 4f, 5b, 5g, Suppl. Fig. 6b,d,e,f, g: The authors should not merge gels that were run separately, i.e. gels should be boxed individually and molecular weight markers should be provided for the separate gels. E.g. the molecular weight marker shown in Fig. 1b belongs to

the left gel, but not to the center or right hand gel. Molecular weights are not indicated in Fig. 4f.

Separate gels have been boxed individually following the reviewer's suggestion. We have tried to label the weight marker for each separate gel (e.g. Fig. 5, presented below for convenience of viewing), however, the figure seems too crowded and messy. Actually, during our figure preparation, separate gels were aligned and the weight markers belong to all separate gels in the same line. We have clarified it in figure legends. In addition, uncropped gels containing molecular makers are provided as a Source Data file. The molecular weights have been labeled in Fig.4f.

- Source Data: Fig. 4e,g are not shown as uncropped gels. Molecular weight markers are missing.

We thank the reviewer for noting this issue for us and we have carefully re-checked our data. Fig.4e and 4g in the Source Data are indeed our original gels. In our experiment, to avoid sample loading error in repeated SDS-PAGE, protein bands were cropped from the same transferred PVDF according their molecular weights and the marker, and then blotted by

specific primary antibodies. We thus seek this reviewer's kind understanding in this matter.

- Fig. 6e: The numbering of the sequences on the left and right of the alignment in Fig. 6e still does not match the sequences shown. E.g. for Nedd4 a sequence stretch from residue 223-347 would refer to 124 aa being displayed in the alignment. What do the numbers in angle brackets refer to? The authors should display the residue numbers of the sequences that are actually shown in the alignment.

The numbers in angle brackets refer to the number of the residues which are not displayed. Following the reviewer's suggestion, we have modified Fig. 6e (presented below for convenience of viewing).

Fig.6e, The primary sequence alignment of L in Nedd4, Nedd4L and Smurf2. The identical residues are coloured in red, and the highly conserved residues are coloured in green. The residues involved in packing with HECT are marked with asterisks.

- Fig. 6c/e: The Smurf2 sequence shown in Fig. 6e corresponds to residues 217-249 (Uniprot identifier Q9HAU4-1, aa 1-748) (and does not start at residue 189, see comment above). This means that, in contrast to Nedd4 and Nedd4L, the linker region in Smurf2 starts 26 aa C-terminal of the WW1 domain and lies directly N-terminal of the WW2 domain. The authors should modify Fig. 6c accordingly.

We thank the reviewer for the suggestion. We have modified Fig. 6c (presented below for convenience of viewing).

Fig.6c, Schematic of Smurf2 domains showing a summary of enzymatic activity derived from the autoubiquitination assay in (d).

- p23-24, protein purification: The authors should either refer to protease 3C or Precision protease or make clear that the two names refer to the same protein. It would be nice to point out that GST-tagged proteins were only used for pull-down assays and all other studies (except substrate ubiquitination) were performed with proteins expressed from pET-32a vectors.

Following the reviewer's suggestion, we have modified the manuscript. We have changed "protease 3C" to "Precision protease". We added the following sentences on page 23 line 15: "Unless otherwise specified, GST-His₆ tagged proteins were used for pull-down assays

and all other studies were performed with Trx-His₆ tagged proteins.”

- p24, lines 5-8 and p25, line 17 are contradictory.

In our experiments, proteins without tags were used for crystallization, while Trx-His-tagged E3s were used in E2-E3 transthiolation assay. We have clarified the description in the revised manuscript on page 23 line 17: “For crystallization, the N-terminal Trx-tagged fragments of recombinant proteins were cleaved by digesting fusion proteins with Prescission protease (50 µg protein with 1 µl protease, Sigma, GE27-0843-01) at 4 °C, and the proteins were purified by another step of SEC.”

- p25, line 18, 22: The concentration of the E3s and DTT is missing.

We have added the concentrations of the E3s and DTT in the revised manuscript (page 25 line 6).

- p26, line 16: The authors should state the buffer conditions in which the proteins were concentrated. Which concentration refers to which protein? Please state the exact concentrations that were used for crystallization.

We have modified the manuscript following the reviewer’s suggestion (page 26 line 5, presented below for convenience of viewing).

“Freshly purified WWP1 2L34HECT was concentrated to 7 mg/ml in Buffer A containing 50 mM Tris (pH 8.0), 500 mM NaCl, 1 mM EDTA and 1 mM DTT. Crystals of 2L34HECT were grown by the hanging-drop vapor diffusion method at 16 °C in a reservoir solution containing 100 mM Tris-HCl pH 7.0 and 15% v/v reagent alcohol, and then were soaked in crystallization solution containing 30% glycerol for cryoprotection. Freshly purified WWP1 L34HECT was concentrated to 15 mg/ml in Buffer A for crystallization. Crystals of L34HECT were grown by the hanging-drop vapor diffusion method at 16 °C in a reservoir solution containing 0.1 M Sodium malonate pH 5.0, 12% w/v Polyethyleneglycol 3350, and then were soaked in crystallization solution containing 25% glycerol for cryoprotection.”

- p26, line 19-21: Which reservoir solution and cryo-condition belongs to which construct?

We have modified the manuscript (see the above point).

- p29, line 11,12: This is not a proper sentence.

Sorry for that, we have modified the sentence on page 29 line 4: “The final model was refined by energy minimization using GROMACS.”

Reviewer #3 (Remarks to the Author):

In their revised manuscript, Wang et. al have made significant improvements in response to the reviewer comments. In their rebuttal letter, the authors have suitably addressed reviewer concerns regarding crystal packing in their structures and a failure to observe the ‘multi-lock’ autoinhibited form of Itch in a previously determined structure (PDB:5XMC). However, much this information has not been incorporated into the revised manuscript and its inclusion

in an additional revision would add to the clarity and scholarliness of the manuscript. I fully support publication of this manuscript in Nature Communications should the following minor issues be addressed:

* In the section titled ‘The multi-lock autoinhibition mode exists in WWP2 and Itch’, the authors should acknowledge that the multi-lock autoinhibited state of Itch (with WW4/HECT interactions) was not observed in the crystal structure of Itch in PDB:5XMC despite the fact that WW3/4 were present in the construct and provide some of the discussion they included in their rebuttal to rationalize this apparent contradiction.

We have revised our manuscript following the reviewer’s suggestion (page 15 line 17, presented below for convenience of viewing).

“We noticed that the WW4-HECT interaction was not observed in the Itch structure (PDB:5XMC) despite the fact that WW3 and WW4 were present in the construct³². Structural analysis revealed that although the WW4-binding “IAY” motif from the N-terminal extension of WWP1 HECT is completely conserved in Itch, there are still some variations in the WW4-HECT packing surface. For example, Trp549HECT which interacts with WW4 in both WWP1 2L34HECT and L34HECT (Fig. 3a and Supplementary 1g), is substituted with Ala in Itch, that may weaken the WW4-HECT interaction. Moreover, when we superimpose WWP1 2L34HECT structure to Itch structure (PDB:5XMC), although WW2-L-HECT in both structures fit very well (Supplementary Fig. 1c), the “Le” site in Itch is occupied by the WW2 domain of a symmetric molecule (Supplementary Fig. 5b). Such competition between a symmetric Itch’ WW2 and WW4 towards HECT during crystallization may be a reason why the WW4-HECT interaction was not observed in the crystal structure of Itch 12L34HECT.”

*In the section ‘Overall structures of the inactive and partially active WWP1’ the authors should include information on crystal packing that they included in their response to the same inquiry from all three reviewers. As it is written now, the described 35 degree rotation of WW4 and 25 degree rotation of L appear to be significant, whereas the authors have concluded in their rebuttal that they are not significant and are likely due to crystal packing. Likewise, in the section ‘The WW2L34-HECT interface’ on page 9 the authors should point out that while one might expect that the network of interactions at the WW4-HECT interface to be different due to the 35 degree rotation of WW4, that the network of contacts is actually similar. They should also include the explanation provided in their rebuttal for how the network of contacts remains similar despite the WW4 rotation.

We thank the reviewer for the suggestion and we have revised our manuscript (page 8 line 15, and page 10 line 1, presented below for convenience of viewing).

“Detailed structural analysis revealed that more crystal contacts exist in 2L34HECT (Supplementary Fig. 1e). E509 from WW4 of 2L34HECT forms a salt bridge with K531 from a symmetric molecule. The main chain of M414 from the N-terminal part of L of 2L34HECT forms a hydrogen bond with N745 of a symmetric molecule. All these crystal contacts do not exist in the L34HECT structure, implying that the rotation of WW4 and the N-terminal part of L in the two structures may arise from crystal contacts.”

“Importantly, although the 35 degree rotation of WW4 and 25 degree rotation of L appear to be significant, the crucial WW4-HECT and L-HECT contacts in the 2L34HECT structure

are preserved in L34HECT (Supplementary Fig. 1g). Our assumption is that the WW4-binding N-terminal extension of HECT is a flexible loop, which allows HECT-bound WW4 to rotate a certain extent upon crystal packing with a symmetric molecule in 2L34HECT; whereas the 25 degree rotation of the N-terminal part of L is the result of the missing WW2 domain and crystal contacts (Supplementary Fig. 1e). Thus, we assume that these crystal contact-induced conformational changes have no functional significance.”

*On page 18, the authors still refer to the Y543E mutant as a phosphomimetic. Apart from harboring a negative charge, glutamate looks nothing like phosphotyrosine as it is far shorter and lacks an aromatic ring. In the third from last sentence on this page, the authors should replace ‘the phosphomimetic mutant’ with a phrase along the lines of ‘introduction of a negative charge at Tyr543 in the form of a Y543E mutation’ and should acknowledge the fact that glutamate and phosphotyrosine are structurally different which introduces potential caveats in interpreting results obtained with the mutant as they pertain to insights on phosphorylation.

We thank the reviewer for the suggestion. We have modified the description in the revised manuscript (page 20 line 8, presented below for convenience of viewing).

“Importantly, introduction of a negative charge at Tyr543 in the form of a Y543E mutation led to elevated ligase activity compared to the WT enzymes (Fig. 3e), suggesting the potential role of Tyr543 phosphorylation in the progression of acute myeloid leukaemia¹⁹, though glutamate and phosphotyrosine are structurally different.”

REVIEWERS' COMMENTS:

Reviewer #2 (Remarks to the Author):

The authors have improved the manuscript and added the requested experiments to substantiate their conclusions. I would consider the manuscript now suitable for publication.